# Variants in the WDR44 WD40-repeat domain cause a spectrum of ciliopathy by impairing ciliogenesis initiation

WDR44 prevents ciliogenesis initiation by regulating RAB11-dependent vesicle trafficking. Here, we describe male patients with missense and nonsense variants within the WD40 repeats (WDR) of WDR44, an X-linked gene product, who display ciliopathy-related developmental phenotypes that we can model in zebrafish. The patient phenotypic spectrum includes developmental delay/intellectual disability, hypotonia, distinct craniofacial features and variable presence of brain, renal, cardiac and musculoskeletal abnormalities. We demonstrate that WDR44 variants associated with more severe disease impair ciliogenesis initiation and ciliary signaling. Because WDR44 negatively regulates ciliogenesis, it was surprising that pathogenic missense variants showed reduced abundance, which we link to misfolding of WDR autonomous repeats and degradation by the proteasome. We discover that disease severity correlates with increased RAB11 binding, which we propose drives ciliogenesis initiation dysregulation. Finally, we discover interdomain interactions between the WDR and $NH_2$-terminal region that contains the RAB11 binding domain (RBD) and show patient variants disrupt this association. This study provides new insights into WDR44 WDR structure and characterizes a new syndrome that could result from impaired ciliogenesis.

Primary cilia are highly conserved microtubule-based hair-like organelles that extend from the plasma membrane of most vertebrate cells and are crucial for normal development and tissue homeostasis[1,2]. The primary cilium serves as a platform for morphogen and growth factor developmental signaling pathways including Hedgehog (Hh) and Wnt[1,3]. Pathogenic variants in genes that are required for primary cilium assembly are associated with a wide range of ciliopathies that share overlapping phenotypic features, including craniofacial, brain, musculoskeletal, and renal abnormalities[4,5].

Assembly of the primary cilium occurs via a multistep process referred to as ciliogenesis. Membrane docking between the mother centriole (MC) distal appendages and preciliary vesicles (PCVs) derived from the endosome recycling compartment and/or the Golgi is a critical early ciliogenesis event requiring small RAB GTPase membrane trafficking regulators[6–8]. PCVs docked to the MC via distal appendage proteins (DAPs) fuse to form a larger ciliary vesicle (CV), which develops into the ciliary membrane that surrounds the growing axoneme. CV membrane assembly is critical for the removal of CP110 and CEP97 from the distal end of the MC to allow axoneme formation[9]. Notably, ciliopathies have been linked with ciliary membrane assembly dysfunction at the MC[10], but not with upstream PCV trafficking stages important for initiating ciliogenesis.

PI3K-Akt signaling regulates ciliogenesis initiation via phosphorylation of the RAB11 effector WDR44 (also known as RAB11BP or Rabphilin-11)[11], encoded by the X-chromosomal gene *WDR44* (OMIM *301070). Under non-ciliating conditions, WDR44 is phosphorylated by Akt at residue serine 342 and serine 344 in the RAB11 binding domain (RBD), causing stronger binding to RAB11 than unphosphorylated WDR44. Under ciliating conditions, PCV trafficking is stimulated by reduced Akt phosphorylation of WDR44, which promotes the switch to a RAB11-FIP3 effector complex with RABIN8[11–13]. Docking of these PCVs to the MC requires the TRAPPII complex protein

✉ e-mail: riceo2@snu.ac.kr; chris.westlake@nih.gov

TRAPPC14, which interacts with both RABIN8 and the DAPs FBF1 and CEP83[13,14]. RABIN8 is also known to directly interact with the DAP CEP164[15]. Consistent with WDR44 functioning as a negative regulator of ciliogenesis initiation, cells depleted of WDR44 by RNAi display trafficking of RABIN8 PCVs to the MC, which stimulates the removal of the CP110 and CEP97 from the MC distal end[11].

WDR44 is a member of the large WD40-repeat (WDR) family, which is characterized by a five to eight bladed β-propeller structure arranged radially around a central channel, giving a donut-like appearance[16,17]. Each blade is formed from highly conserved repeats that fold into a 4-stranded β-sheet with strands labeled A-D. The 'donut hole' region of the WDR structure serves as a binding pocket or interaction hub. WDR44 is predicted to have a seven-bladed β-propeller at the very carboxy end of the protein, and there is currently no known function attributed to this domain. In addition to negatively regulating ciliogenesis, WDR44 has also been linked to other vesicular membrane trafficking processes including cell surface transport of transferrin, E-cadherin, CFTR, and matrix metalloproteinases-14[18,19]. These functions are attributed to interactions with RAB11 and/or the membrane trafficking regulators VAPA and GRAF2, which all have discrete binding sites in the $NH_2$-terminal region of WDR44 that are not found in other WDR proteins.

In the present study, we describe seven missense and one nonsense variants in the WDRs of WDR44 identified from eleven male patients displaying a wide range of cognitive impairment and variable congenital anomalies associated with primary cilium dysfunction. Our findings support expression of WDR44 variants in patients' cells and suggest disease severity is associated with altered protein abundance. WDR44 containing ciliopathy-related missense variants are subject to enhanced proteasomal degradation, which can be explained by protein misfolding resulting from variants affecting the highly conserved blades of the WDR β-propeller structure. We further show that WDR44 variant disease and protein abundance effects are modeled in zebrafish embryos. Importantly, we find ciliogenesis initiation and ciliary signaling are impaired in patient-derived fibroblasts from the more severely affected patients. Expression of WDR44 variants in human cells and zebrafish embryos further demonstrates dysregulated ciliogenesis initiation as a contributing factor in patient disease. Using biochemistry and cellular localization approaches, we discovered WDR44 variants have a gain-of-function (GOF) in binding to RAB11, albeit independent of effects on Akt phosphorylation of the RBD, which can explain why ciliogenesis is impaired. Furthermore, we show the WDR44 COOH-terminal fragment containing the WDR interacts with the $NH_2$-terminal fragment containing the RBD and patient variants disrupt this interdomain interaction. Together, our findings suggest that patient variants in the WDR domain cause a previously uncharacterized pleiotropic ciliopathy-related disorder associated with disrupted WDR44 interdomain interactions important for regulating RAB11 binding that controls ciliogenesis initiation.

## Results

### Identification of *WDR44* variants

In a research program dedicated to investigating individuals with unsolved genetic diseases, trio-based exome sequencing (ES) was performed for a male (III:1 of family 1) with mild intellectual disability (ID) and multiple congenital anomalies including microcephaly, unilateral multicystic kidney, musculoskeletal abnormalities, and craniofacial dysmorphism (Fig. 1a–f, S1a–e, Table S1), phenotypes that overall suggest a possible underlying ciliopathy disorder.

Stepwise filtering of ES analysis failed to identify pathogenic or likely pathogenic variants in any known disease-causing gene. Based on our hypothesis of a possible ciliopathy-related phenotype in the index subject, we next prioritized variants in genes known to have pivotal functions in cilia development and function. Accordingly, we retained a maternally inherited variant [GenBank: NM_019045.5, c.2291C > T p.(S764F)] in *WDR44* that was absent in the gnomAD database (Table S2). Sanger sequencing confirmed that it was hemizygous in the proband and heterozygous in his mother who was healthy, while four maternal uncles were unaffected. This missense variant results in the substitution of a phylogenetically conserved amino acid localized in the 5th WDR (WD5) in the COOH-terminal region of the protein (Fig. 1g, h). A comprehensive analysis looking at all genes and considering all possible mechanisms of inheritance did not identify stronger candidate variants (Supplementary Data 1). WDR44 was recently listed as an OMIM gene (creation date 02/15/2022) and has some constraint for missense variation (gnomAD Z-score 2.95) and is intolerant for loss of functon variants (gnomAD pLI 1.0).

This finding led us to investigate whether other patients harboring novel and/or ultrarare WDR44 variants in the WDR domains present similar phenotypes to the identified male. We screened for WDR44 variants in genomic datasets and searched publicly available patient cohorts (*see Methods*). This effort led us to identify 10 additional patients from 8 unrelated families displaying a neurodevelopmental phenotype and variable associated anomalies. In all subjects, data analysis excluded the presence of other pathogenic or likely pathogenic variants in known neurodevelopmental disorder (NDDs)-related genes. Sanger sequencing confirmed segregation of the variants with the phenotype within these families including three affected males of family 4 (Fig. S1a).

Overall, we identified six additional missense variants [c.1943A > G p.(D648G), c.2003T > C p.(L668S), c.2005G > A p.(D669N), c.2344G > T p.(G782C), c.2516A > G p.(H839R), c.2519A > G p.(N840S)] and one nonsense variant [c.2197C > T p.(R733*)]. All variants were maternally inherited with the exception of c.2344G > T p.(G782C) (family 2) and c.2003T > C p.(L668S) (family 5) which were de novo variants (Fig. S1a). Genotyping analysis supporting true maternity and paternity of families 2 and 5 is available in Table S3. Notably, the p.L668S variant was observed in two unrelated patients (family 5 and 6), being maternally inherited in family 6. All but one variant are absent from the gnomAD database: the c.2519A > G p.(N840S) has an allele frequency of 0.00001783, however, it has never been reported in hemizygous state (Table S2). All missense variants affect highly conserved residues in the WD repeat domain (Fig. 1h) and are predicted to be deleterious according to multiple prediction tools (Table S2) with the exception of p.(N840S), which we classified as a variant of uncertain significance. The REVEL score for this variant falls within the tolerable range (0.17), p.(L668S) is borderline damaging (0.63), and all other variants are predicted to have damaging consequences. Variants classification is provided in Table S2. The nonsense variant in exon 16 introduces a premature stop at codon 733 of the encoded protein. This is expected to lead to an absent protein and/or a protein that is truncated at residue 733 in WD5 (Fig. 1g).

### Phenotypic spectrum associated with WDR44 variants

All 11 subjects were males and presented with neurodevelopmental issues: eight had global developmental delay and were eventually diagnosed with either mild (n = 5) or moderate (n = 3) ID, and three individuals had learning disabilities (Table S1). The most common craniofacial dysmorphism included upslanting palpebral fissure (n = 8), long and/or smooth philtrum (n = 8) and thin upper lip vermilion (n = 9) (Fig. 1a–c, S1b and Table S1). Musculoskeletal abnormalities were predominant in our cohort and included joint hypermobility (n = 8), brachydactyly (n = 6), II-III toe syndactyly (n = 4), congenital hip (n = 3), knee (n = 2) and elbow (n = 1) dislocation, talipes equinovarus (n = 3), and scoliosis (n = 3) (Fig. 1d, S1d–e, Table S1). Notably, II-III toe syndactyly was observed in both unrelated p.L668S patients. Variable renal abnormalities were noticed in three patients, including unilateral multicystic kidney disease in one; and nephritis, and kidney hypoplasia resulting in renal failure in two patients (Fig. 1e and Table S1). Five

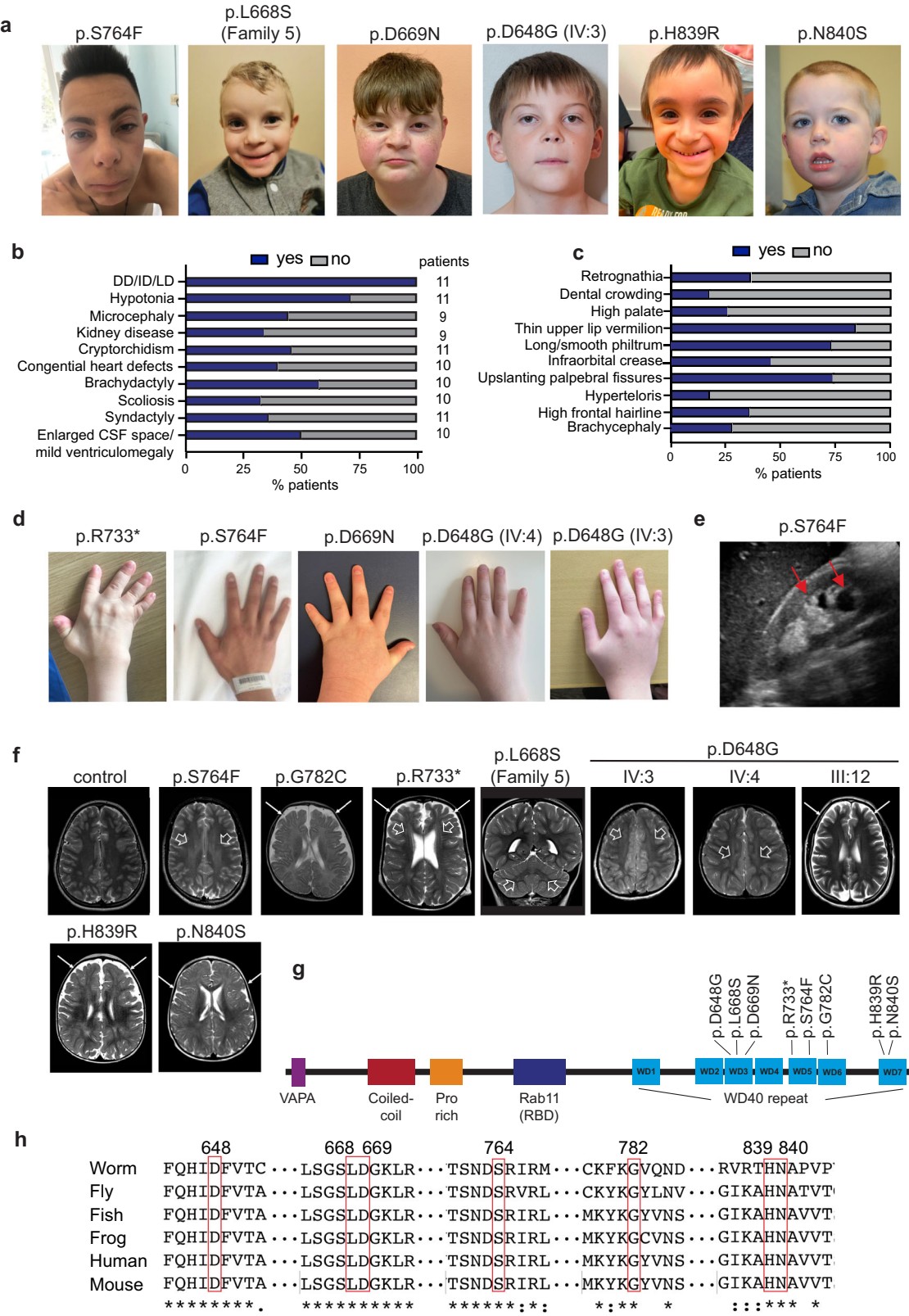

individuals were found to have cryptorchidism that required orchidopexy. Four subjects had congenital heart defects. Neurological evaluation revealed hypotonia in eight subjects and secondary microcephaly in four. Brain MRI scans were available in nine patients and was abnormal in all, including mild enlargement of the subarachnoid spaces with white matter volume reduction and ventricular enlargement ($n = 5$) and anterior commissure hypoplasia ($n = 5$),

simplification of the gyral pattern and faint T2/FLAIR signal alterations of the cerebral and cerebellar white matter ($n = 5$), and corpus callosum hypoplasia ($n = 3$) (Fig. 1f, S2). With the exception of some findings such as the white matter changes, most of the clinical features together were suggestive of a ciliopathy-related spectrum. A summary of all other clinical features of affected individuals is presented in Table S1 and Fig. 1b.

**Fig. 1 | Clinical assessment of WDR44 variants. a** Photographs of subjects with WDR44 variants demonstrate common craniofacial features including high frontal hairline, upslanting palpebral fissure, smooth philtrum and thin upper lip vermilion in all patients with the exception of subject III:2 of family 9 harboring the c.2519A > G p.(N840S) who does not display major dysmorphism. Patients harboring the c.2291C > T p.(S764F) (Family 1) and c.1943A > G p.(D648G) (Family 4) variants also show long philtrum and mildly webbed neck. (**b**, **c**) Bar graph showing the distribution of the clinical and craniofacial features. Blue: presence of ciliopathy-related features. Gray: absence of ciliopathy-related features. Analysis for 11 patients (**c**). **d** Photos of hands showing digital abnormalities including brachydactyly in all showed patients, and camptodactyly of fifth fingers for patient with c.1943A > G p.(D648G) (family 4) and ulnar deviation in the patient harboring the c.2197C > T p.(R733*) variant (family 3). **e** Kidney ultrasound of patient with c.2291C > T p.(S764F) variant shows parapelvic cysts (red arrows). **f** Neuroimaging features of the affected subjects with normal control for comparison. Brain MRI with axial T2-weighted images showing mild simplified gyral pattern and faint T2-weighted hyperintensity of the deep fronto-parietal white matter (white empty arrows) in subject III:1 of Family 1 and IV:4 and IV:3 of Family 4. Brain MRI with axial T2 weighted images showing mild enlargement of the subarachnoid spaces (thin arrows) with white matter volume reduction and ventricular enlargement in subjects II:1 of Family 2, III:12 of Family 4, III:2 of Family 8, and III:2 of Family 9. **g** The schematic represents the WDR44 protein and its domains. The position of variants is indicated in the WDR domain. **h** Multiple sequence alignment analysis of WDR44 wild-type protein across species. The numbers represent the position of conserved amino acid residues that are altered in the patients. Source data are provided as a Source Data file. DD/ID/LD developmental delay/intellectual disability/learning disability, CSF cerebrospinal fluid.

## WDR44 patient variants affect protein stability

*In-silico* programs predict missense variants to be deleterious and affect protein stability with the exception of p.N840S (Table S2). To investigate the protein stability related pathogenicity of WDR44 variants, fibroblasts were cultured from patients harboring the p.D648G (IV:4), the p.S764F, and the p.N840S variants and parents or unrelated controls. Immunoblotting analysis demonstrates that the protein abundance of variants p.D648G and p.S764F is strongly reduced compared to control fibroblasts (Fig. 2a), and the p.S764F expressed at significantly higher levels than the p.D648G (Fig. 2a). In contrast, the variant p.N840S protein level in fibroblasts was not significantly different compared to the father's cells protein expression. Examination of *WDR44* wild-type and variants mRNA levels in controls and patient fibroblasts showed no difference in the transcription of the gene with missense variants indicating that altered protein levels were a post-transcriptional effect (Fig. 2b). Variant effects on missense protein abundance could also be observed by examining overexpressed GFP-tagged WDR44 (GFP-WDR44) wild-type and variants in 293T cells (Fig. 2c). Notably, all missense variants showed lower protein levels compared to the wild-type protein, with L668S and N840S variants having the least effect on protein expression (Fig. 2c). In contrast, the nonsense p.R733* variant mRNA levels in patient fibroblasts was significantly reduced compared to controls suggesting expression may be affected by nonsense mediated decay (NMD) (Fig. 2b). However, GFP-tagged R733* variant was expressed at similar level as the wild-type tagged protein (Fig. 2c). Together, these results demonstrate that exogenously expressed patient variants have variable effects on WDR44 protein abundance.

To determine if missense variants are degraded by the proteasome, patient fibroblasts were treated with MG132. p.D648G and p.S764F variants protein levels were increased by MG132 treatment by greater than 2-fold (p.D648G, $p = 0.0014$; p.S764F, $p = 0.0004$) (Fig. 2d), suggesting variants are degraded by the proteasome. The p.N840S variant ($p = 0.037$) and the wild-type protein levels were less affected by MG132 treatment (Fig. 2d). Similar results were observed with overexpressed GFP-WDR44 wild-type and missense variants in 293T cells (Fig. 2e). Notably, chloroquine did not affect WDR44 protein levels indicating that protein degradation does not occur via lysosomal degradation (Fig. 2e). Together, these results suggest that patient missense variants in the WDR destabilize WDR44 protein structure, resulting in proteasomal degradation.

To further test the impact of missense variants on WDR folding, we expressed COOH-terminal fragments (WDR44 COOH-GST, 477–913) in 293T cells. Reduced protein fragment levels were observed for the missense variants compared to the wild-type protein, similar to that observed with full-length proteins and indicative that WDR folding is likely affected by the amino acid substitutions (Fig. 2f). Interestingly, a homology-based 2D proteomap of the WDR shows that all variants except p.N840S are located on the WDR blades 3, 5, or 6 (Fig. S3a). Apart from p.N840S, the other missense variants lie in strands of the blades and, therefore, may be more likely to affect the folding of the beta-sheet repeats. This positioning of missense variants is supported by the AlphaFold2-predicted 3D structural model of the COOH-terminal WDR domain (residues 480 to 913) (Fig. 2g). To further investigate the effect of patient variants on the WDR structure we performed molecular dynamics (MD) simulations with and without patient variants. Overall, our MD simulations were too short (500 ns) to observe large-scale conformational changes of the core WD40 domain (Fig. S3b, c); however, local conformational changes of mutated residues were observed for D669N, S764F, G782C, and H839R, indicating instabilities and potential aberrant protein folding of the WDR blades (Fig. 2h and S3d).

## WDR44 variants affect ciliogenesis

Given the known function of WDR44 in ciliogenesis, we investigated ciliation in patient missense variant fibroblasts. We evaluated cilia levels by immunostaining control and patient fibroblasts under low ciliating conditions (with serum) and high ciliating conditions (serum starvation) (Fig. 3a). p.D648G and p.S764F patient fibroblasts had little ciliation when grown in serum compared to controls. However, following serum starvation p.S764F patient cells had similar ciliation to the controls, while p.D648G patient fibroblasts could ciliate but significantly less than in controls. Variant protein stability was still affected following serum starvation and, therefore, is not associated with improved ciliation (Fig. S4a). In contrast, p.N840S patient fibroblasts did not display any difference in cilia number compared to the parental fibroblasts in either condition (Fig. 3a). To determine if reduced ciliation can be rescued by wild-type WDR44, we examined ciliation after stably expressing GFP and GFP-WDR44 in fibroblasts (Fig. 3b). Cilia levels were restored in the p.S764F patient fibroblasts under low ciliating conditions, but not in the p.D648G fibroblasts (Fig. 3c). Thus, we can conclude that overexpressing the wild-type protein can rescue p.S764F variant effects on ciliation. We also examined cilia length and did not observe significant differences in cilia length between controls and p.D648G, p.S764F, and p.N840S patient fibroblasts under serum starvation conditions (Fig. S4b). However, the cilia length observed in the p.N840S patient fibroblast in low ciliating conditions was significantly shorter than that observed in his parents fibroblast (Fig. 3d). To further examine the impact of WDR patient variants on ciliation, we expressed Myc-tag WDR44 (Myc-WDR44) wild-type and variants in 293T cells and evaluated ciliogenesis under serum fed conditions (Fig. S4c). Overexpression of all WDR44 variants significantly reduced cilia length and did not affect ciliation levels, which may be explained by the presence of endogenous WDR44 protein. Together, our results indicate that WDR44 variants can affect ciliation.

## Impaired ciliogenesis in patient fibroblasts affects Hedgehog signaling

Ciliogenesis defects associated with ciliopathy affect developmental signaling pathways such as Hh. Consequently, we examined

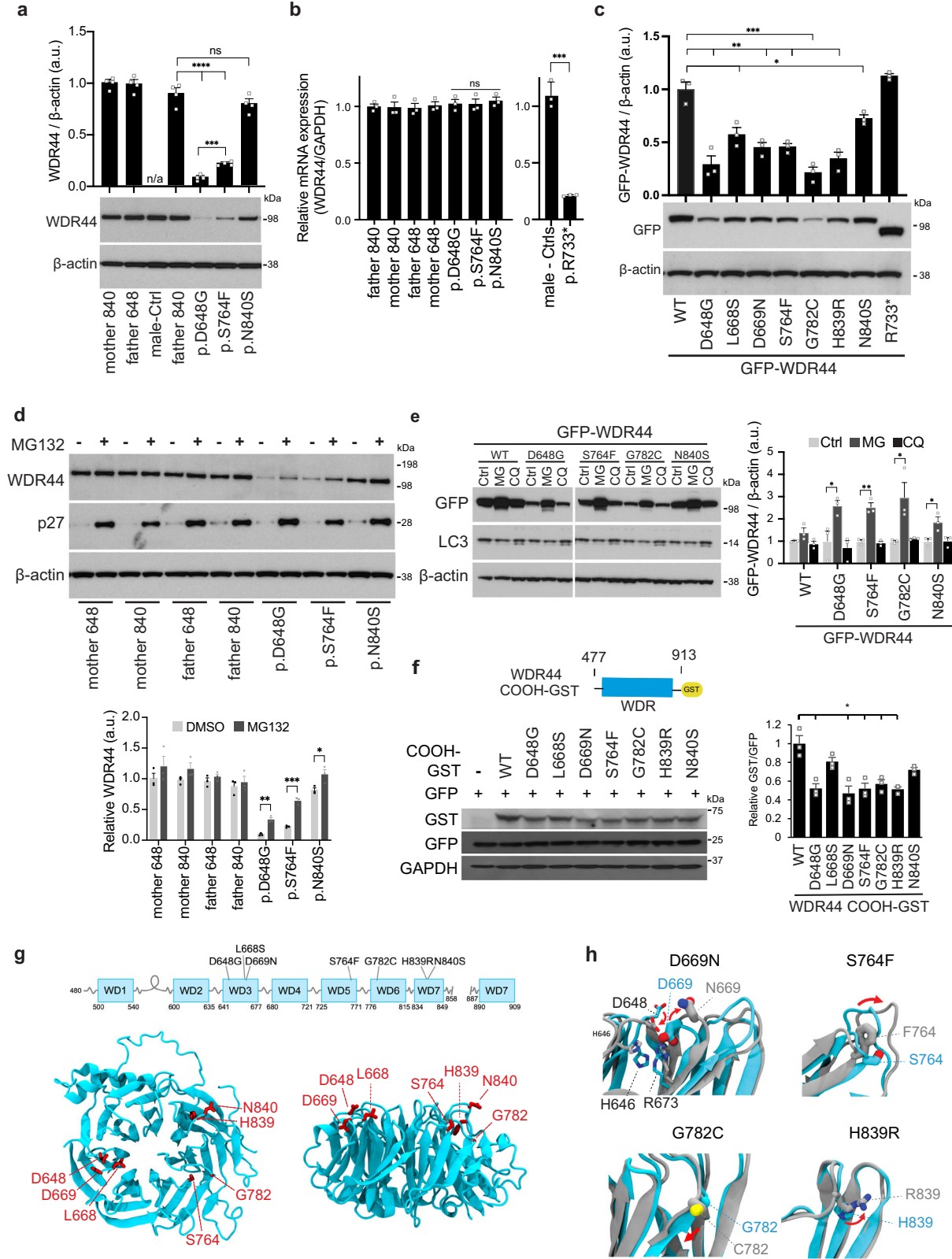

Hh signaling targets *GLI1* and *PTCH1* transcripts in human fibroblasts with and without pathway activation by using the sonic hedgehog agonist (SAG) (Fig. 3e, f). Consistent with lower ciliation observed in p.D648G and p.S764F patient fibroblasts grown with serum we observed reduced *GLI1* and *PTCH1* transcript levels compared to controls with SAG treatment (Fig. 3e, f). SAG induced

*GLI1* and *PTCH1* transcript levels were increased in p.D648G and p.S764F patient fibroblasts in high ciliation conditions, although the *GLI1* levels were still significantly lower than in control fibroblasts which could be associated with differences in ciliation observed between these cells (Fig. 3a). Hh signaling in p.N840S patient fibroblasts was not significantly affected compared to controls in

**Fig. 2 | WDR44 variants affect protein stability and reduce expression.**
**a** Immunoblotting analysis of WDR44 and β-actin in lysates from control (matched and unmatched parents) and *WDR44* variant patient fibroblast cells. Proteins band relative intensity is indicated by the graph (*above*). Statistical comparisions with father 840 or p.D648G are shown. $P \le 0.0001$ (p.D648G, p.S764F), 0.0002 (p.S764F). **b** WDR44 mRNA relative expression was determined by real-time RT-PCR in control and WDR44 variant patient fibroblasts from 3 independent mRNA isolations. **c** Immunoblotting analysis of transiently expressed GFP-WDR44 wild-type and variants in 293T cells post 48 h of transfection. Proteins band relative intensity is indicated by the graph (*above*). Statistical comparisons with wild-type (WT) are shown. $P = 0.0108$ (L668S), 0.0233 (N840S), 0.0026 (D648G), 0.0027 (D669N), 0.002 (S764F), 0.002 (H839R), 0.0008 (G782C). **d** Immunoblotting analysis of proteins from the lysates of 24 h vehicle control (-) or 1μM MG132 (+) treated fibroblast cells. p27 is used as a positive marker of proteasome inhibition. Proteins band relative intensity is indicated by the graph (*below*). Statistical comparisons between DMSO vehicle control and MG132 are shown. $P = 0.0365$ (p.N840S), 0.0014 (p.D648G), 0.0004 (p.S764F). **e** Immunoblotting analysis of transiently expressed GFP-WDR44 wild-type or variants in 293T cells treated with vehicle control (Ctrl) or 1μM MG132 (MG) for 16 h or 10μM Chloroquine (CQ) for

24 h. Protein band relative intensity is indicated by the graph (*right*). Statistical comparisons between Ctrl and MG are shown. $P = 0.0355$ (D648G), 0.0422 (G782C), 0.029 (N840S), 0.0018 (S764F). **f** Immunoblotting analysis of COOH-terminal domain of wild-type or variants in 293T cells 48 h after transfection. GFP was used as an internal control of transfection. Schematic above represents the COOH-terminal region of WDR44 used for the analysis. Proteins band relative intensity is indicated by the graph (*right*). $P = 0.0144$ (D648G), 0.0105 (D669N), 0.0132 (S764F), 0.0207 (G782C), 0.0230 (H839R). **g** Predicted structure of the WDR44 WDR domain for residues 480–858 and 887–913 modeled from the AF predicted structure (AF-Q5JSH3). Residues mutated in patients are shown in red. **h** Structure alignment of WDR44 WDR wild-type (cyan) and patient missense variant (grey). The patient variant structures were built from the structure described in (**g**) followed by 500 ns MD simulation to assess conformational changes. Patient variants shown to have the largest conformational changes based on RMSD/residue analysis in Fig. S3d. Red arrows show change in position of patient variants from wild-type. Mean ± s.e.m. from three (**b**–**f**) or four (**a**) independent experiments. Unpaired two-tailed *t*-test; *$P < 0.05$, **$P < 0.01$, ***$P < 0.001$, ****$P < 0.0001$, non-significant (ns). Source data are provided as a Source Data file.

low and high ciliating conditions (Fig. 3e, f). These results demonstrate that more severely affected WDR44 patient cells have affected Hh signaling, important in development and associated with ciliopathy.

### Patient WDR44 variants cause developmental defects in zebrafish

To determine if we can model WDR44 ciliopathy-related disease, we used zebrafish for functional studies after expressing the human variants. Human and zebrafish Wdr44 share 69% sequence identity and the missense patient variants are completely conserved (Fig. 1h). *Wdr44* shows broad expression across the zebrafish embryo at 1 day post fertilization (dpf) by in situ hybridization chain reaction (HCR) analysis, as was previously reported[20], and remains expressed in ciliated organs at 2 dpf (Fig. S5a, S5f). Embryos were treated with *wdr44* morpholinos (MOs) to reduce expression of the endogenous Wdr44 protein and were validated by examining a GFP reporter RNA containing the 5′ UTR target site of the MOs (Fig. S5b). Notably, neither MOs show obvious morphogenic defects compared to uninjected controls at 2, 3, or 6 dpf (Fig. 4a–h, S5c, d). With the exception of N840S, the WDR44 variants showed low or undetectable protein levels by western blotting in lysates of zebrafish morphants (Fig. S5e), however, dot blot experiments confirmed all patient variants were expressed at lower levels than the wild-type WDR44 (Fig. 4a). Together, these results suggest protein stability of WDR44 variants is also affected in zebrafish.

Next, we examined the effects of WDR44 variants on zebrafish development (Fig. 4a–h). Compared to morphants co-injected with WDR44 wild-type mRNA, significant developmental defects were observed in morphants expressing WDR44 L668S, D669N, S764F, G782C, H839R, and R733* variants (Fig. 4b–h) including craniofacial phenotypes (head dysmorphism), body curvature and heart development defects. Furthermore, the zebrafish phenotypes presented various degrees of severity consistent with the phenotypic spectrum and the disease-causing character of the identified human variants. Noteworthy, injections with D648G or N840S had no significant effects on zebrafish development at three dpf. Finally, D669N, G782C, H839R, N840S, and R733* variants caused significant distended pronephric tubules, a characteristic of cystic pronephros, compared to WDR44 wild-type (Fig. 4h). Consistent with these phenotypes glomerular dilations were observed with D669N, S764F, G782C, H839R, N840S, and R733* variants (Fig. 4i). Kidney cysts were identified in the human patient with the p.S764F variant (Fig. 1e). Together, the majority of WDR44 variants recapitulate the WDR44-associated human features in zebrafish.

### WDR44 variants cause ciliogenesis defects in zebrafish embryos

We investigated ciliation in zebrafish morphants expressing WDR44 variants by specifically examining immotile cilia in the neuromasts, and both motile and immotile cilia in the pronephros, and olfactory placodes (Fig. 5a–c). At 3 dpf, we show that *wdr44* morphants embryos with and without WDR44 wild-type expression had similar ciliation as uninjected controls (Fig. 5a–c, S5g). Consistent with these results, ciliation was unaffected in these organs in embryos treated with a CRISPR-Cas9 gRNA to knockout *wdr44* (Fig. S5h). In contrast, morphants expressing WDR44 variants, except for D648G, L668S and N840S, caused defective ciliation in the neuromasts, olfactory placodes and pronephric ducts. These latter results correlate with the observation of pronephric cysts in morphants expressing five of the variants at 6 dpf (Fig. 4h). Interestingly, N840S treated embryos showed some cystic pronephros at 6 dpf yet ciliation at 3 dpf appeared normal. One explanation for this result is that subtle defects in ciliation at 3 dpf may result in cyst formation later. Together, these findings are consistent with ciliation defects in patients fibroblasts and 293T cells suggesting that various WDR44 variants exert a GOF in repressing ciliogenesis.

### WDR44 variants affect ciliogenesis initiation by preventing mother centriole uncapping

To examine if ciliogenesis initiation is affected by WDR44 variants we investigated MC uncapping in control and patient fibroblasts by immunostaining cells for CP110. Examination of CP110 at the MC and daughter centriole (DC) showed that p.D648G and p.S764F cells maintained CP110 at the MC at significantly higher levels than in controls under low ciliating conditions (Fig. 6a). Under high ciliation conditions, p.S764F fibroblasts had normal MC uncapping, whereas p.D648G uncapping was significantly reduced (Fig. 6a), results consistent with ciliation levels observed in these patient cells (Fig. 3a). p.N840S cells uncapping in serum fed and serum starved conditions was not different than controls. Together, these results demonstrate that more severe disease is associated with affected ciliogenesis initiation in patient fibroblast cells.

To further investigate WDR44 variant effects on ciliogenesis initiation we generated a RPE-1 cell line lacking WDR44 (RPE-1 WDR44 KO) (Fig. S6). Ablation of WDR44 promotes ciliogenesis in three different clones (E1, F4, G2) compared to control RPE-1 (Ctrl Cas9) cells under low ciliating conditions (Fig. 6b). F4 clone WDR44 knockout (KO) was verified by genotyping (Fig. S6) and this cell line was used to examine the effects of WDR44 variant expression on MC uncapping under serum starvation conditions (Fig. 6c). Transient expression of wild-type GFP-WDR44 did not prevent MC uncapping compared to

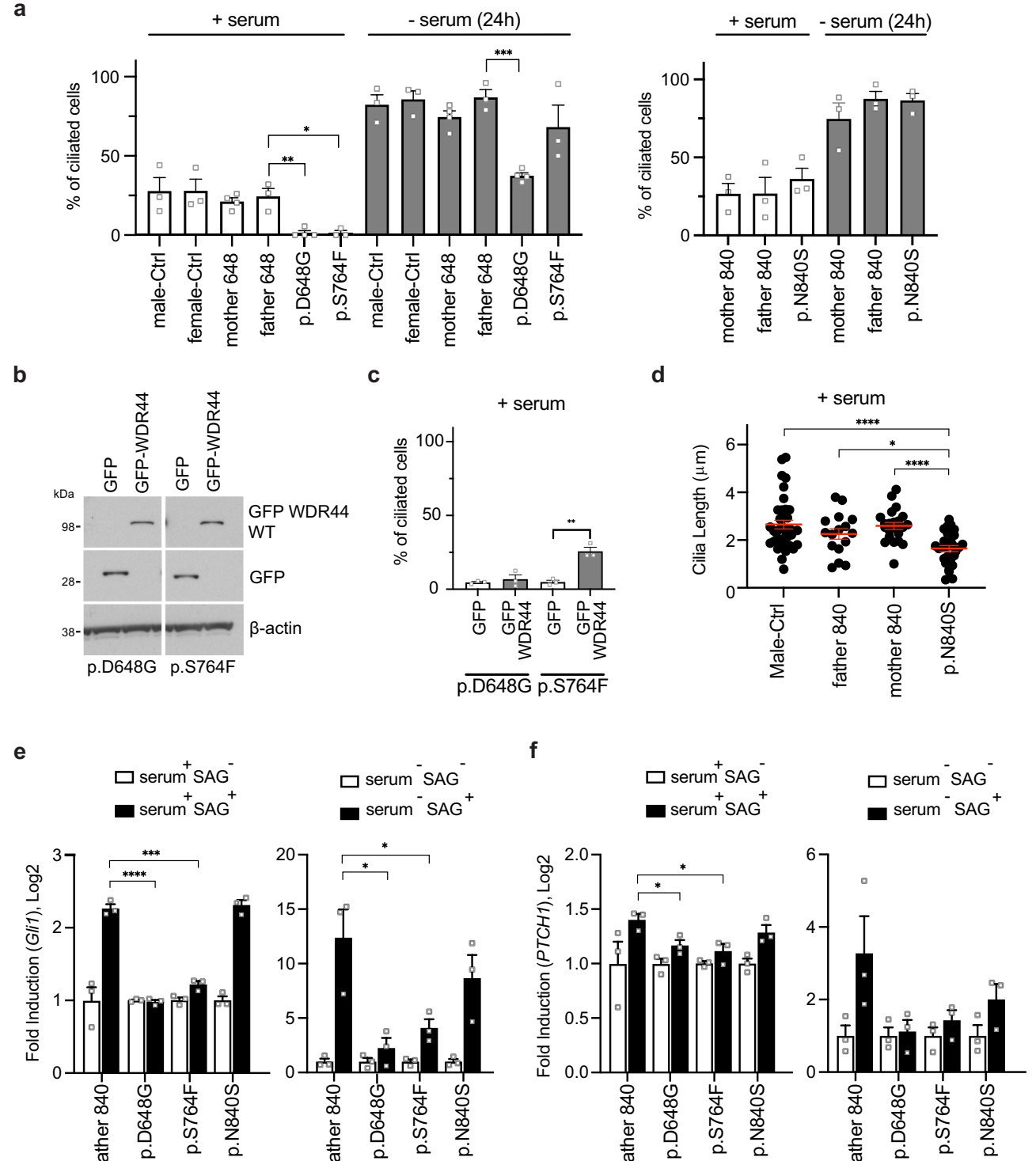

**Fig. 3 | Patient fibroblasts have affected ciliation and Hedgehog signaling.**
**a** Quantification of ciliation in control and WDR44 variant (p.D648G, p.S764F, p.N840S) patient fibroblast fed with 10% serum (+serum) or starved (−serum) for 24 h, followed by anti-Arl13b, anti-CEP164, and anti-CP110 antibodies staining. >150 cells were counted from three or more independent experiments. Statistical comparisons with father 648 are shown. +serum: $P = 0.012$ (p.S764F), 0.0039 (p.D648G), −serum (24 h): $P = 0.0001$ (p.D648G). **b** Immunoblotting analysis of GFP or GFP-WDR44 wild-type and β-actin from lysates of patient fibroblasts. Probed for GFP and β-actin antibodies. **c** Ciliation was quantified in p.D648G and p.S764F patient fibroblast rescued with GFP control or GFP-WDR44 wild-type. >150 cells were counted from three independent experiments. $P = 0.0024$ (p.S764F).

**d** Quantification of cilia length in fibroblast from p.N840S and controls that were stained as in (**a**). Statistical comparisons with p.N840S. $P = 0.0131$ (father 840), <0.0001 (Male-Ctrl, mother 840). (**e**, **f**) Hedgehog signaling was analyzed in parental control and patient fibroblasts. *GLI1* (**e**) and *PTCH1* (**f**) transcript fold induction determined by real-time RT-PCR in fibroblast fed with 10% serum (serum+) or starved (serum−) in the presence (SAG+) or absence (SAG−) of SAG for 24 h. Statistical comparisons with father 840 SAG+ are shown. **e** serum+: $P = 0.0001$ (p.S764F), <0.0001 (p.D648G), serum−: $P = 0.0211$ (p.D648G), 0.0375 (p.S764F). **f** serum+: $P = 0.03$ (p.D648G), 0.025 (p.S764F). Mean ± s.e.m. from three independent experiments. Unpaired two-tailed *t*-test; *$P < 0.05$, **$P < 0.01$, ***$P < 0.001$, ****$P < 0.0001$. Source data are provided as a Source Data file.

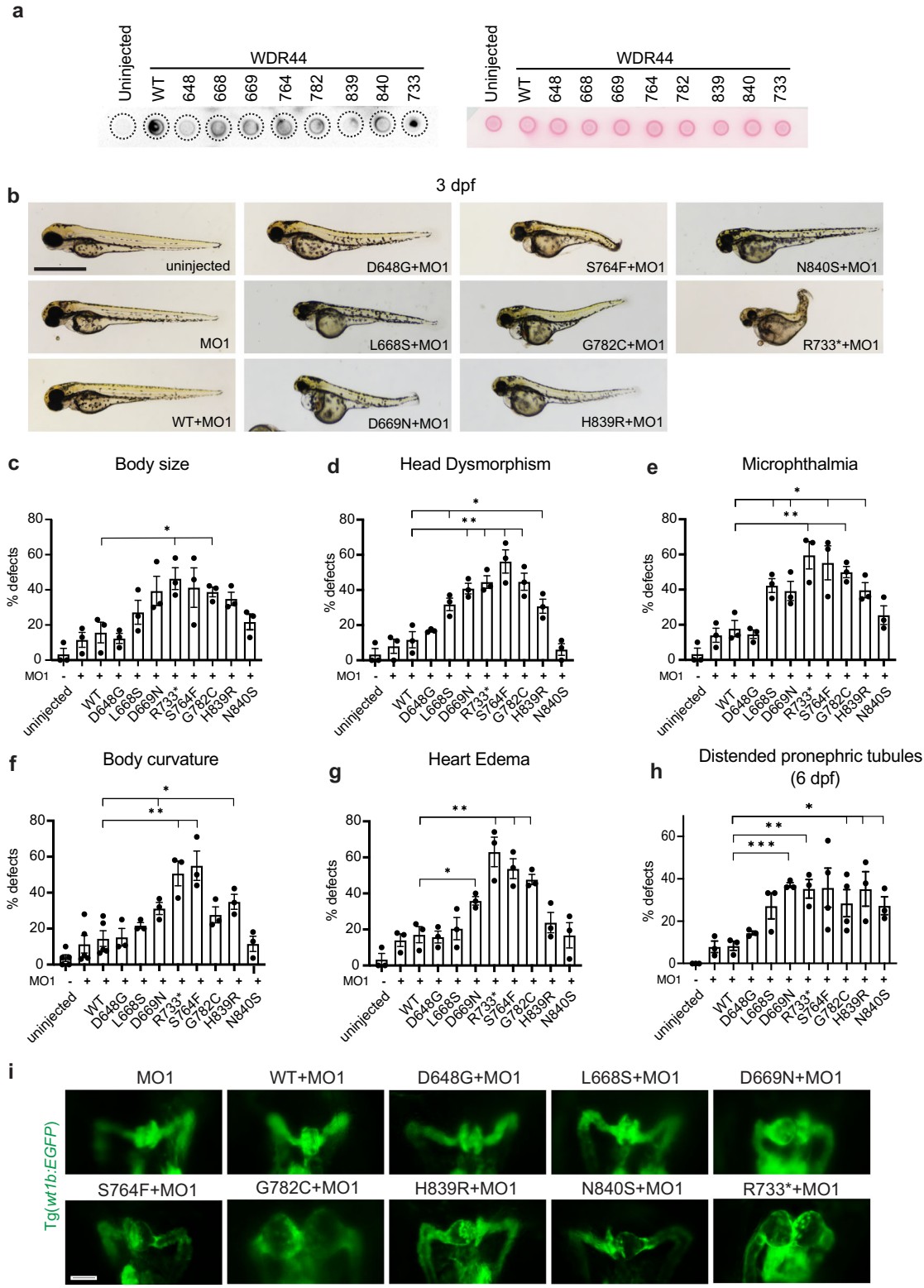

**Fig. 4 | WDR44 variants affect zebrafish embryonic development. a** Dot blot analysis of WDR44 wild-type and variants expressed in zebrafish morphants (*left*). Protein loading was demonstrated by Ponceau staining of dot blot (*right*) probed with WDR44 antibodies. **b** Bright-field (BF) image analysis of zebrafish embryos injected with *wdr44* morpholino and rescued with mRNA of human WDR44 wild-type or variants. Scale bar, 1 mm. (**c**–**h**) Quantification of developmental defects: body size, head dysmorphism, microphthalmia, body curvature, heart edema at 3 dpf, and distended pronephric tubules at 6 dpf. Statistical comparison with WT are shown. **c** *P* = 0.0229 (R733*), 0.0236 (G782C). **d** *P* = 0.0261 (L668S), 0.0062 (D669N), 0.0049 (R733*), 0.0052 (S764F), 0.0079 (G782C), 0.0355 (H839R).

**e** *P* = 0.0157 (L668S), 0.0404 (D669N), 0.0099 (R733*), 0.0251 (S764F), 0.0046 (G782C), 0.0260 (H839R). **f** *P* = 0.0379 (D669N), 0.0032 (R733*), 0.0028 (S764F), 0.0204 (H839R). **g** *P* = 0.0204 (D669N), 0.0080 (R733*), 0.0067 (S764F), 0.0044 (G782C). **h** *P* = 0.0003 (D669N), 0.0053 (R733*), 0.0480 (G782C), 0.0331 (H839R), 0.0163 (N840S). **i** Representative images of 3 dpf Tg(*wt1b*:EGFP) embryos injected as in (**b**) showing developing pronephros. Scale bar, 50 μm. Mean ± s.e.m. from three independent experiments imaging >60 embryos (**c**–**h**) or >10 embryos (**i**). Unpaired two-tailed *t*-test; *P < 0.05, **P < 0.01, ***P < 0.001. Source data are provided as a Source Data file.

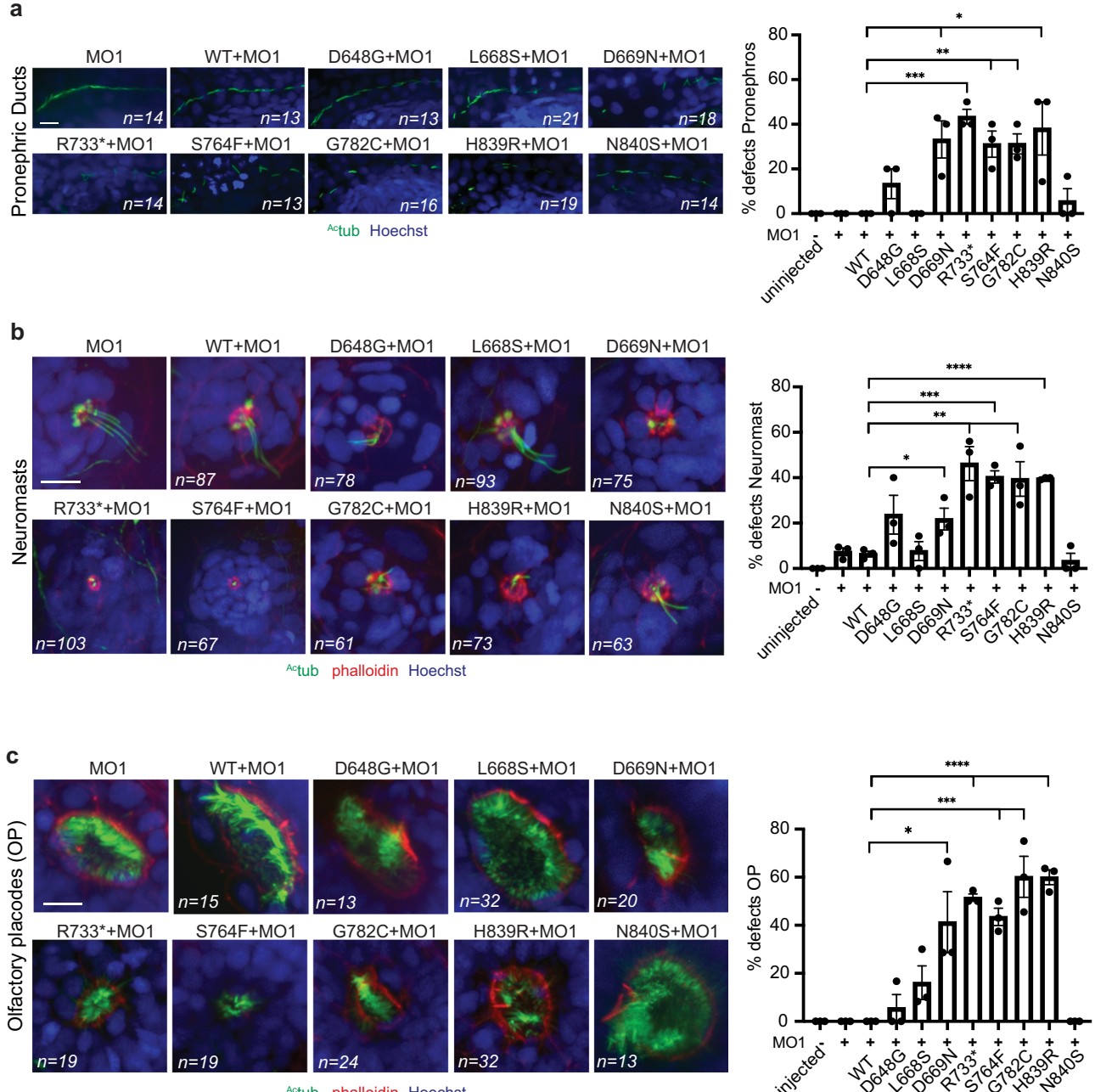

**Fig. 5 | WDR44 variants reduce ciliogenesis in zebrafish embryos.** Quantification (*right*) of ciliation in pronephric ducts (**a**), neuromasts (**b**), and olfactory placode (**c**) in zebrafish embryos at 3 dpf. Embryos were injected with *wdr44* morpholino with or without mRNA of human WDR44 wild-type and variants, followed by staining with anti-acetylated tubulin antibody and Hoechst. Phalloidin staining was performed in (**b, c**). Represented immunofluorescence microscopy (IFM) (*left*) images. Scale bars, 10 μm. Statistical comparison with *wdr44*MO + WT are shown.

**a** $P = 0.0161$ (D669N), 0.0002 (R733*), 0.0061 (S764F), 0.0023 (G782C), 0.0328 (H839R). **b** $P = 0.1155$ (D648G), 0.0360 (D669N), 0.0064 (R733*), 0.0003 (S764F), 0.0129 (G782C), <0.0001 (H839R). **c** $P = 0.0309$ (D669N), <0.0001 (R733*), 0.0003 (S764F), 0.0021 (G782C), $P < 0.0001$ (H839R). Mean ± s.e.m. from three independent experiments and total number of embryos imaged are indicated on images. Unpaired two-tailed *t*-test; *$P < 0.05$, **$P < 0.01$, ***$P < 0.001$, ****$P < 0.0001$. Source data are provided as a Source Data file.

GFP alone, while all the WDR44 missense variants significantly reduced CP110 loss (Fig. 6c). Together, these results indicate that all missense WDR44 variants can affect ciliation initiation upstream of MC uncapping. The GFP-WDR44 R733* nonsense variant had less than 50% (47.7 ± 15.9%) CP110 uncapping compared to nearly 75% observed with the wild-type protein (73.7 ± 4.5%), albeit this difference was not statistically significant ($p > 0.2$). Notably, at higher expression levels the nonsense variant was more toxic to the cells than the wild-type, and therefore only cells with lower expression of this variant could be analyzed, which could affect the results obtained.

## WDR44 variants affect RAB11 localization and display stronger binding to RAB11

WDR44 negatively regulates ciliogenesis initiation by binding to RAB11 and blocking PCVs trafficking to the MC[11–13]. Therefore, we examined whether the patient variants in the WDR domain affect interaction with RAB11. We performed co-expression co-immunoprecipitation studies using Myc-WDR44 wild-type or variants and GFP-RAB11A in 293T cells (Fig. 7a). Strikingly, all the WDR44 variants bound significantly more strongly to GFP-RAB11A than the wild-type protein (Fig. 7a). Myc-N840S bound significantly less strongly to GFP-RAB11A than the other

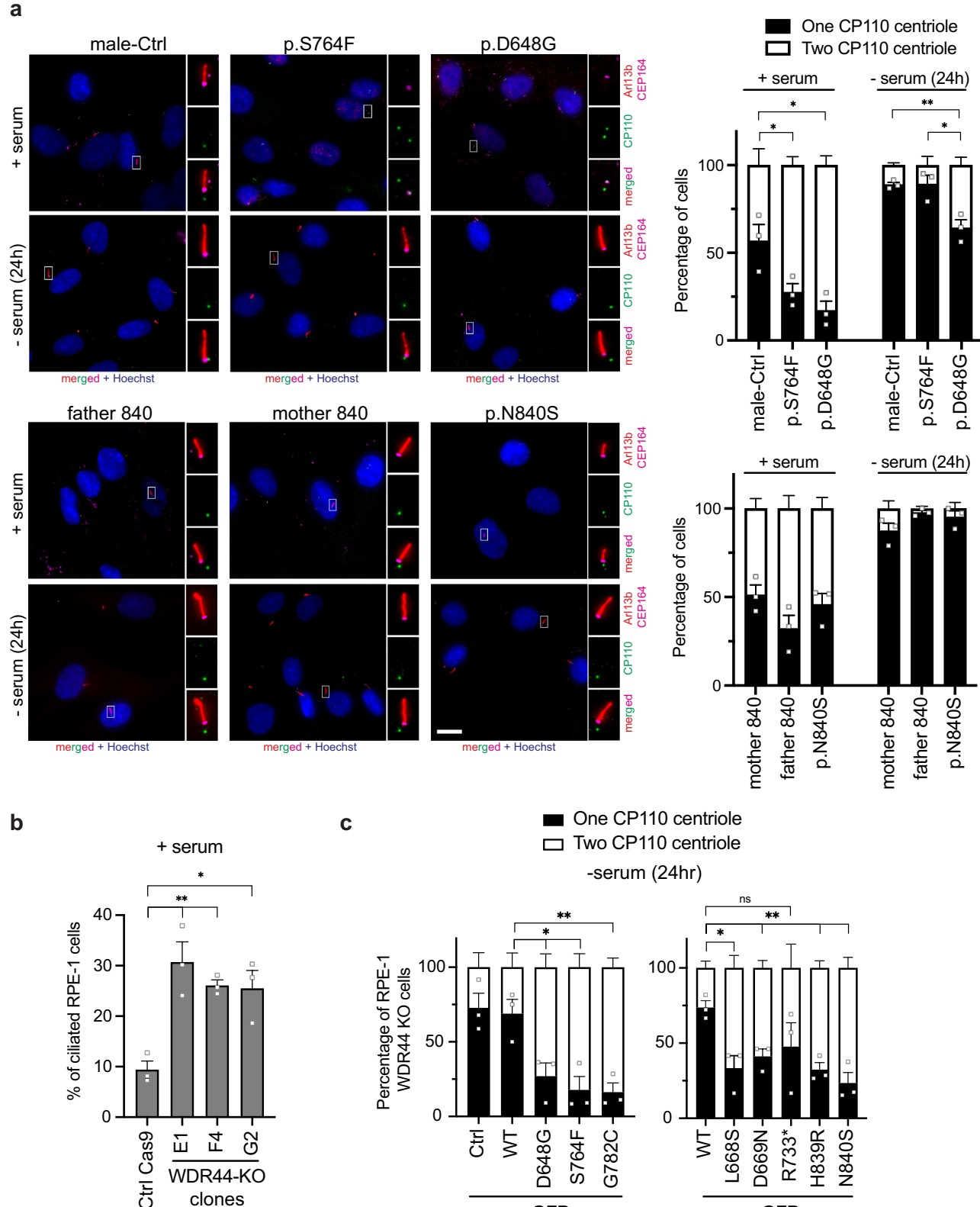

Myc-tagged missense variants, with the exception Myc-L668S. Interestingly, Myc-R733* binding to GFP-RAB11A was only found to be significantly less than that observed for D648G and S764F. Attempts to detect endogenous RAB11-WDR44 interactions by immunoprecipitation studies in patient cells were unsuccessful, however we could demonstrate RAB11 expression is unaffected by WDR44 variants (Fig. S7). Endogenous RAB11 binding to transiently expressed GFP-

WDR44 wt and variant proteins showed a similar trend in RPE-1 WDR44 KO cells (Fig. 7b). In addition, interaction with VAPA was stronger with the GFP-tagged D648G and R733* variants (Fig. 7b) suggesting that other WDR44 interactions could also be affected by variants. Together, these results provide a molecular explanation for impaired ciliogenesis initiation caused by WDR44 variant enhanced binding to RAB11. Indeed, more severe disease is associated with variant protein

**Fig. 6 | WDR44 variants reduce ciliogenesis initiation. a** Quantification (*right*) of MC uncapping (denoted as one CP110 centriole on daughter centriole) and capped MC (denoted as two CP110 centriole on both mother and daughter centriole) in the patient and control fibroblast fed with 10% serum (+serum) or starved (−serum) for 24 h. Cells were stained with anti-Arl13b, anti-CEP164, and anti-CP110 antibodies. Nuclei were stained with Hoechst 33342. Represented IFM images (*left*). Scale bars, 10 μm. >100 cells were counted from three independent experiments. Statistical comparisions with male-Ctrl or p.S764F are shown. +serum: *P* = 0.0499 (p.S764F), 0.0211 (D648G), −serum (24 h): *P* = 0.0211 (p.D648G), *P* = 0.0065 (pD648G). **b** Percentage of ciliation determined by immunostaining RPE-1 control Cas9 and

WDR44 KO clones with anti-Arl13b and anti-CEP164. >150 cells were counted from three independent experiments. Statistical comparisons with Ctrl Cas9 are shown. *P* = 0.0148 (G2), 0.0081 (E1), 0.0011 (F4). **c** Quantification of MC uncapping as in (**a**) in the RPE1 WDR44 KO F4 cells transiently expressing GFP control or GFP-WDR44 wild-type or variants for 48 h. >60 cells with similar GFP expression were counted from three independent experiments. Statistical comparisons with GFP-WDR44 wild-type (WT) are shown. *P* = 0.0329 (D648G), 0.0178 (S764F), 0.0131 (L668S), 0.01 (G782C), 0.0084 (D669N), 0.0032 (H839R), 0.0039 (N840S). Mean ± s.e.m.. Unpaired two-tailed *t*-test; *P* < 0.05, **P* < 0.01, non-significant (ns). Source data are provided as a Source Data file.

abundance and affinity changes for RAB11, and potentially other interacting proteins.

We next examined the effects of WDR44 variants on the subcellular localization of RAB11 in RPE1 WDR44 KO cells. Consistent with our previous report[11], GFP-WDR44 wild-type localizes to the cytoplasm and on small punctate structures that partially colocalizes with RAB11 membrane vesicles (Fig. 7c), which are enriched in the pericentriolar region of the cell[21]. In contrast, the GFP-WDR44 variants, except N840S, showed less cytoplasmic localization with nearly all the detectable protein colocalized with RAB11 positive vesicles (Fig. 7c). Interestingly, RAB11 vesicles were noticeably larger in cells expressing these GFP-WDR44 variants. Moreover, in the case of D648G, L668S, S764F, G782C, and R733* expression, RAB11 showed less accumulation in the perinuclear region and was found towards the periphery of cells. Thus, these results suggest that WDR44 variants associated with more severe disease can affect the cellular distribution of RAB11 due to their higher affinity for the protein.

### Patient variants affect interdomain interaction

Our findings support the conclusion that WDR44 variants have GOF in suppressing ciliogenesis initiation via sequestering RAB11 to prevent PCV trafficking. This raises questions about how variants in the COOH-terminal region of WDR44 affect RAB11 binding as previous work showed this domain is dispensable for interaction with this small GTPase[19]. Because Akt phosphorylation of WDR44 at Ser342 and Ser344 is important for WDR44-RAB11 interaction, we examined whether this is affected by patient variants. Akt phospho-WDR44 (pAKT WDR44) levels correlated with protein abundance in control and variant expressing human fibroblasts and 293T overexpressed proteins (Fig. 8a-b, S8). Furthermore, a comparison of Akt phosphorylation of WDR44 variants at low and high ciliating conditions demonstrated that pAKT WDR44 levels were affected by serum starvation as expected (Fig. 8a, b). Thus, stronger binding between the WDR44 variants and RAB11 was not due to changes in Akt phosphorylation of WDR44.

We next considered whether the WDR domain could influence RAB11 binding by affecting interdomain associations within WDR44. To test this hypothesis, we performed coimmunoprecipitation studies in 293T cells coexpressing the NH$_2$-terminal fragment containing the RBD (HA-1-408 AA) and the COOH-terminal WDR domain (GST-COOH domain, 477–913 AA) of WDR44 (Fig. 8c). Consistent with our hypothesis, the HA-tagged NH$_2$-fragment of WDR44 could pull-down the wild-type WDR domain (Fig. 8c). Strikingly, the WDR domains carrying patient variants showed negligible or strongly reduced binding to the NH$_2$-terminal fragment. These results suggest that patient variants affect interdomain interactions between the WDR and the NH$_2$-terminal regions containing the RBD. Interestingly, disease severity also correlated with disruption in WDR44 interdomain interactions as the N840S COOH-terminal fragment binding to the NH$_2$-fragment was significantly less affected than with the other variants in comparison to the wild-type fragment. Together, our findings support a model whereby patient missense variations in the β-propeller structure of the WDR destabilize the protein to varying degrees and alter interdomain interactions that

could influence RAB11 binding, which in turn affects ciliogenesis initiation.

## Discussion

In the present study, we describe a novel pleotropic developmental disorder associated with ciliopathy-related features due to X-linked *WDR44* variants. Based on comprehensive in vitro and in vivo experiments we propose a molecular mechanism of disease wherein WDR44 variants within the WDR result in a GOF in negative regulation of ciliogenesis through increased RAB11 binding activity. Interestingly, we find that more severe disease causing missense variants have a destabilizing effect on the protein likely resulting from structural defects in the WDR. Moreover, WDR patients variants disrupt interdomain interactions that are associated with regulating binding to RAB11. While our studies support a GOF mechanism, we cannot rule out that loss of function (LOF) of WDR44 contributes to the ciliopathy-like phenotype observed, because protein expression data was only available for a limited number of patients. Future experimentation with additional patient cells, as well as assessing neurological function and learning/memory in knockout animal models, will be important for further defining patient disease mechanisms.

The patients that we identify with variants in the WDR44 WDR display an array of overlapping developmental abnormalities affecting craniofacial, musculoskeletal and brain development. Ciliary dysfunction can commonly lead to these abnormalities. In particular, we found high prevalence of skeletal defects in our cohort including brachydactyly, syndactyly, and scoliosis[22,23]. Our patients also display neurodevelopmental features in line with the plethora of developmental and brain disorders linked to ciliopathies[24]. Notably, half of our cohort and most of the zebrafish larvae expressing different variants present with microcephaly, which has been linked to cilium assembly/disassembly as well as disruptions in centrosome or centriolar function associated with proliferation defects[25,26].

Hh and Wnt signaling play an important role in craniofacial, musculoskeletal and brain development[23,27,28]. Disruption in cilia-dependent signaling of these pathways is well known to lead to a spectrum of patterning defects in digit formation and within the facial midline consistent with what is observed in WDR44 patients. Abnormal forebrain development due to Hh signaling impairment[29] leads to simplified gyral pattern, callosal abnormalities, and enlarged ventricles which were observed in some of our patients[30,31]. Interestingly, half of the cohort showed overlapping cerebral and cerebellar white matter changes (Fig. 1f, S2a, b). Although this neuroimaging finding is not typically reported in ciliopathies, it is notable that primary cilia are present in oligodendrocyte progenitor cells and Shh signaling is crucial for the generation of myelinating oligodendrocytes from these precursors[32]. However, it is also possible that this neuroimaging pattern might be due to disruption of an extra-ciliary function of WDR44 that is yet unknown. Our results in patient fibroblasts show that reduced ciliation in p.D648G and p.S764F variant cells is associated with lower Hh signaling, while shorter cilia in the p.N840S patient cells did not significantly affect Hh signaling (Fig. 3e−f), although shorter cilia have been linked to ciliopathies[33]. These results are consistent with the p.D648G and p.S764F variants causing ciliopathy-related

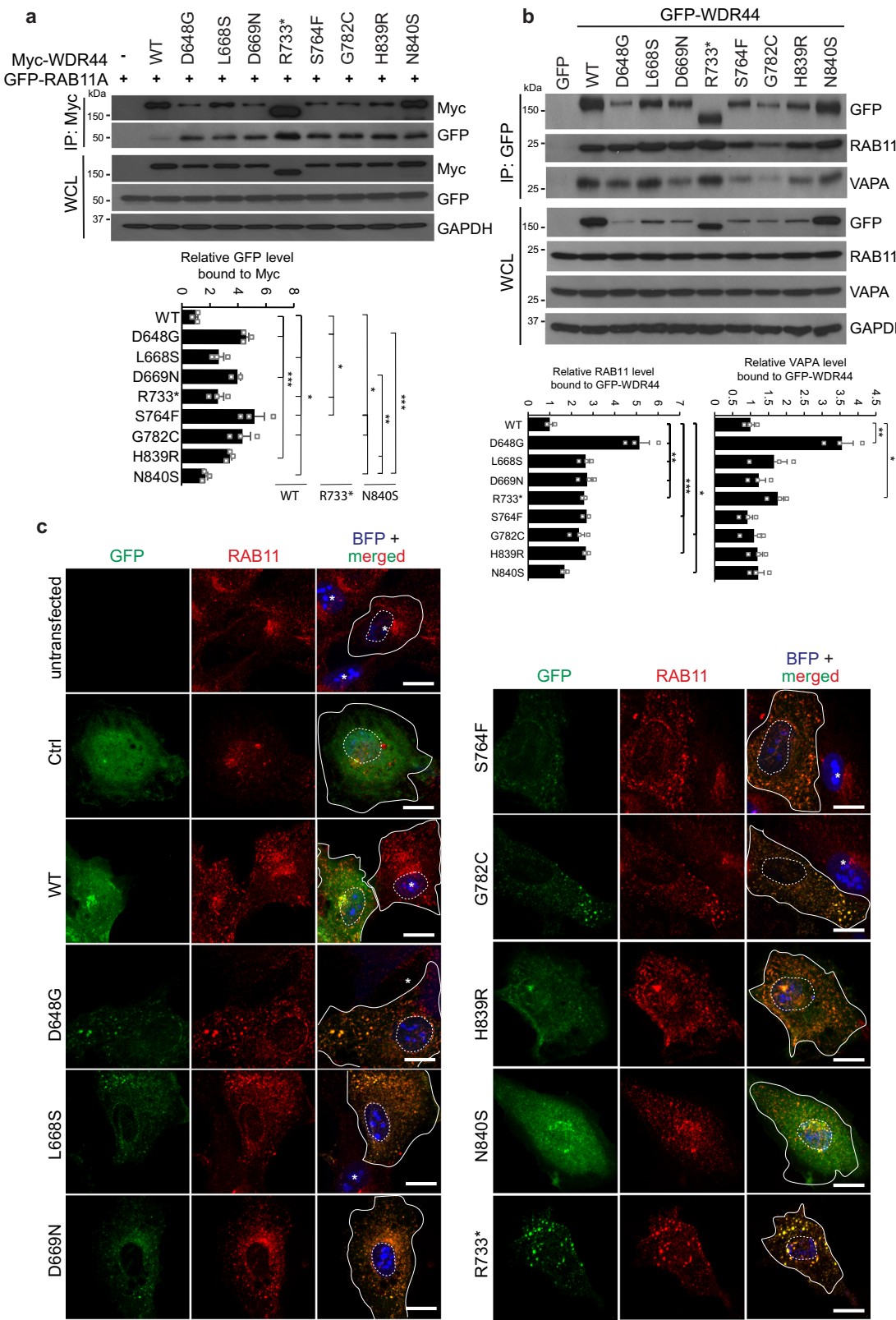

disease. As other WDR44 variants can reduce ciliogenesis when exogenously expressed in human cells and zebrafish, it is plausible that ciliary signaling could be similarly affected in these patients. In comparison to the p.N840S variant, our studies with other patient missense variants showed reduced protein stability, higher RAB11 binding, disrupted WDR44 interdomain interactions, and/or strongly impaired ciliogenesis (Fig. 8d). Based on these findings and the lack of

overlapping craniofacial features and congenital anomalies observed in other subjects, we cannot rule out that the p.N840 variant might have a mild ciliopathy or non-functional consequence and this patient phenotype may be due to a yet unknown genetic etiology. Along this same line of thinking, differences in patient genetic background may explain why ciliation in the p.D648G variant fibroblast could not be rescued by expressing the wild-type GFP-WDR44 as compared to the

**Fig. 7 | WDR44 patient variants enhance binding to RAB11 and cause RAB11 redistribution in cells. a** Coimmunoprecipitation analysis of coexpressed GFP-RAB11A and Myc-WDR44 wild-type or variant in 293T for 48 h. Myc-tagged proteins were immunoprecipitated from 293T lysate and probed with GFP, Myc, and GAPDH. The plot (*below*) shows the relative co-immunoprecipitated levels of GFP-RAB11A bound to Myc-WDR44 wild-type or variants from three independent experiments. Statistical analysis comparing wild-type (WT), R733* and N840S are shown vs WT: $P = 0.0269$ (L668S), 0.0459 (R733*), 0.0236 (S764F), 0.0226 (G782C), 0.0261 (N840S), 0.0002 (D648G), 0.0009 (D669N), 0.0002 (H839R); vs R733*: $P = 0.0459$ (WT), 0.0221 (D648G), 0.0448 (S764F); vs N840S: $P = 0.0261$ (WT), 0.0333 (S764F), 0.0356 (G782C), 0.0018 (D669N), 0.0011 (H839R), 0.0004 (D648G). **b** Immunoprecipitation analysis of GFP-WDR44 wild-type or variants transiently expressed in RPE1 WDR44 KO F4 cells for 48 h. GFP proteins were immunoprecipitated from lysate and probed with GFP, RAB11, VAPA, and GAPDH antibodies. Blots are represented from three independent experiments. The plots (*below*) shows the relative co-immunoprecipitated levels of RAB11 and VAPA bound to GFP-WDR44 wild-type or variants. RAB11: $P = 0.0183$ (G782C), 0.0135 (N840S), 0.0086 (D648G), 0.0022 (L668S), 0.0065 (D669N), 0.0023 (R733*), 0.0005 (S764F), 0.0009 (H839R); VAPA: $P = 0.0229$ (R733*), $P = 0.0087$ (D648G). **c** IFM images of RPE1 WDR44 KO F4 cells transfected with GFP or GFP-WDR44 wild-type or variants. Cells express BFP and are immunostained with RAB11. Plasma membrane and nuclei are indicated by solid and dotted lines, respectively. Representative images from 20–25 cells are shown. White asterisk denotes untransfected cells. Scale bars, 10 µm. Mean ± s.e.m. from three independent experiments. Unpaired two-tailed *t*-test; *$P < 0.05$, **$P < 0.01$, ***$P < 0.001$. Source data are provided as a Source Data file. WCL whole cell lysate.

p.S764F patient cells. However, these results could also be due to the D648G variant having enhanced interaction with other proteins including VAPA. Finally, it should also be noted that patient fibroblasts tested in this study were generated in different clinics and thus cell culturing conditions and passage number could be associated with the ciliation levels observed.

Severely affected patients display other classical ciliopathy phenotypes that could be related to GOF in impairing ciliogenesis initiation. The finding of renal cysts, non-immune nephritis, and renal hypoplasia leading to chronic kidney disease in three of our patient cohort is consistent with the phenotypic spectrum observed in renal ciliopathies[34,35]. The observation of unilateral renal cysts observed in patient p.S764F has also been reported in Joubert syndrome and Bardet-Biedl Syndrome[36]. The human renal phenotype was modeled in the zebrafish expressing WDR44 variants showing defective ciliation in the pronephros, which is consistent with the observed grossly distended pronephric tubules and dilated glomeruli that are characteristic of cysts in this organ (Fig. 4h, i). Monitoring these patients for kidney disease over time is important as late onset presentations of cystic kidney disease have been reported for several ciliopathies[34]. It is noteworthy that renal cysts, polydactyly, hip anomalies, and talipes equinovarus have been previously described in a male patient with 1.9 Mb deletion in Xq24 that encompasses the first exon of the *WDR44* gene[37]. However, this work did not establish whether *WDR44* was responsible for the observed phenotype as three other genes were also deleted.

Ciliopathy associated congenital heart defects were also observed in three of our patients, and heart edema was observed in zebrafish embryos expressing WDR44 variants[38,39]. Another well observed abnormality in ciliopathy patients is cryptorchidism[40], which was reported in five of our patients. Overall, our clinical and experimental findings support WDR44 variants as the cause of a pleiotropic disorder in eight of nine cohort families involving many of the organ systems typically affected in ciliopathies through a cilia-linked mechanism. This observed WDR44-related disorder might therefore belong to the continuously expanding list of second-order ciliopathies, i.e., diseases that are caused by pathogenic variants in genes encoding protein that are not localized within cilia but that have a role in cilium formation and function[1].

Biochemistry analysis, *in-silico* prediction analysis, and MD simulations of the WDR structure suggest that more severe disease associated GOF WDR44 missense variants affect the folding of the protein causing instability leading to proteasomal degradation, yet these structural changes can enhance RAB11 binding affinity. The impact of these factors on the severity of disease can be seen by comparison of the more moderately affected individuals with the p.D648G variant and the more severely affected individual with p.S764F variant. While both variants showed similar strong enhancement in RAB11 binding, the D648G protein appeared to be less stable, bind VAPA more strongly, and is associated with higher levels of ciliation than the variants affecting the WDR blade 5 (Figs. 2a, 4a, 5a–c and 7b). In the case of the p.R733* truncation variant the mRNA appears to be sensitive to NMD

(Fig. 2b), yet the protein can be detected in exogenous expression studies in human cells and zebrafish (Figs. 2c, 4a and 7a, b) even though it is predicted to lack the final two blades of the β-propeller (Fig. S3a). Interestingly, this truncation occurs at the very end of one of the D strands and therefore it may be possible this protein forms a stable five-bladed WDR β-propeller structure. Notably, the p.R733* variant displays more severe disease than the missense variants in the patients and zebrafish. Exogenous expression in fish embryos and human cells suggests this protein is associated with higher expression levels than missense variants, which could result in a more pronounced GOF on RAB11-dependent ciliogenesis initiation. Unfortunately, further studies with p.R733* protein from patient fibroblasts were not possible at this time and therefore we cannot rule out LOF as a mechanism of disease.

To understand how missense variants in the COOH-terminal WDR domain influence RAB11 interaction with the more NH$_2$-terminal RBD, we hypothesized that interdomain interactions are important for WDR44 ciliogenesis functioning. Indeed, we show the WDR domain immunoprecipitated with the NH$_2$-terminal RBD-containing domain and remarkably, missense variants associated with more severe ciliopathy-related disease prevent this interaction. Thus, this finding links WDR domain function to influencing RAB11 binding affinity and the regulation of ciliogenesis initiation. Interestingly, the position of amino acid residues that are mutated in WDR44 patients are near the surface of the protein known to form a binding pocket important for protein-protein interactions in other WDR-containing family members (Fig. 2g)[16,17]. Short MD simulations and other structural prediction programs (Table S2) indicate missense variants on the blades of β-propeller can affect the local WDR structure and, therefore, could impact blade folding, which as we demonstrate, impacts the interdomain interaction and ultimately leads to the mutant protein being degraded by the proteosome. Notably, the p.N840S variant is located in a linker between the D- and A-strand and structure predictions indicate this variant is tolerated, perhaps because it is less likely to affect the blades of β-propeller than the other missense variants affecting blade strands (Fig. S3a). One possibility is WDR44 interdomain interactions influence RAB11 binding to the RBD by affecting an open and closed conformation; open conformation with stronger RAB11 binding and a closed conformation where interdomain interactions involving the WDR reduces RAB11 binding affinity. This model is supported by a previous report indicating only a small fraction of WDR44 binds to the membrane via RAB11 and it was theorized that the larger cytosolic fraction has a different conformation associated with the accessibility of the RBD[19]. Indeed, our observation that more severe ciliopathy-associated WDR44 variants show stronger association with enlarged RAB11 membrane compartments supports this theory (Fig. 7c). Although RAB11 interaction with the WDR alone has not been reported[19], an alternative explanation is that the WDR surface binding pocket could be involved more directly in RAB11 interaction together with the RBD. Both of these interaction models could be influenced by Akt phosphorylation in the RBD. Still another possibility exists, where the WDR44 WDR binding pocket interacts with an unknown protein which can influence binding to

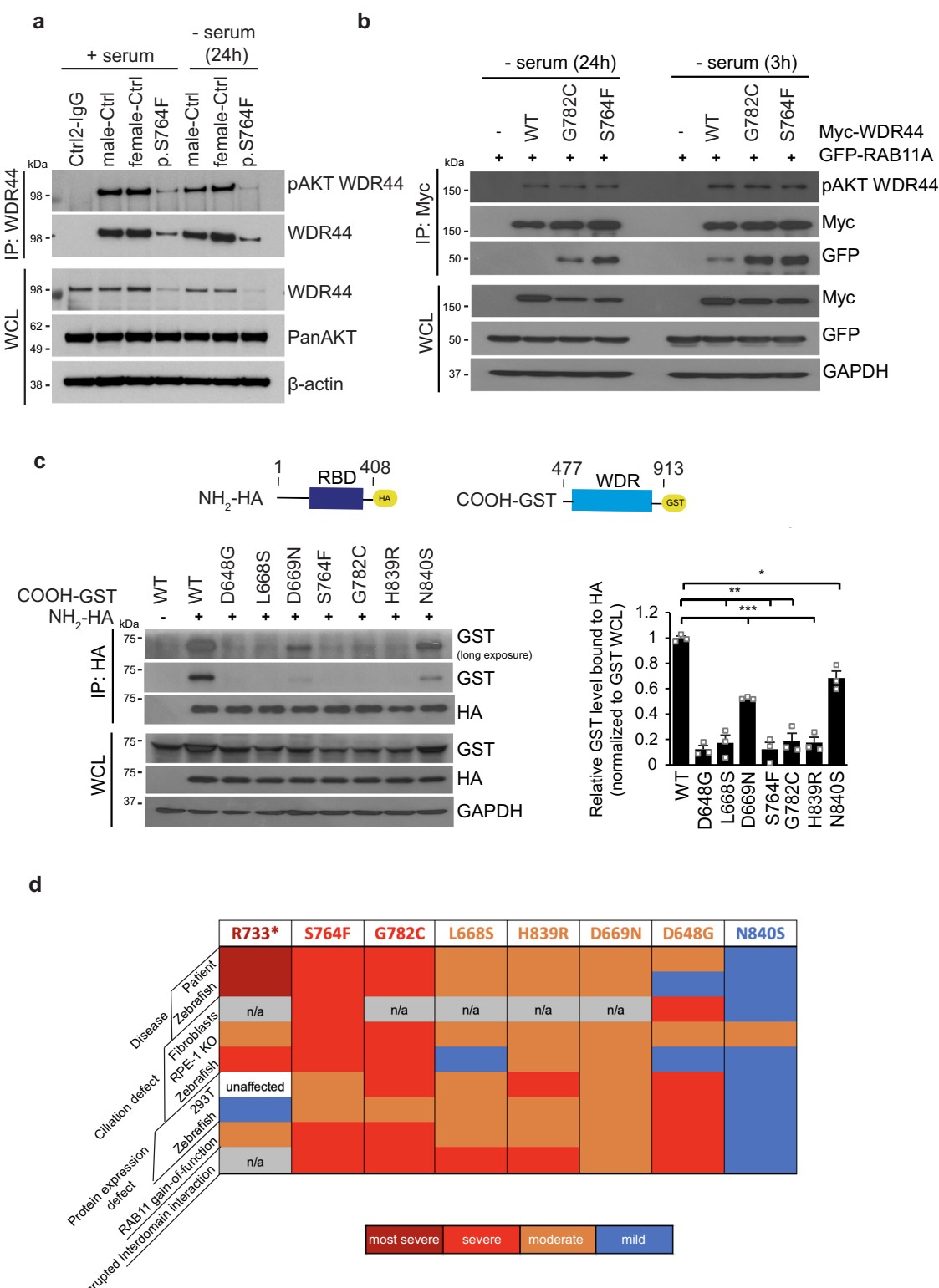

RAB11. Future studies aimed at defining the WDR44 structure will be important to understand how this protein regulates ciliogenesis initiation and causes disease when mutated in patients.

In conclusion, our study unveils a novel pleiotropic developmental disorder due to *WDR44* variants resulting in a ciliopathy spectrum associated with impaired ciliogenesis initiation, and possibly a more complex disease that also affects non-ciliary related WDR44 function. Identification and characterization of additional patients with variants in the WDR and other domains will help to increase understanding of this unique disorder caused by a negative regulator of ciliogenesis and provide new insights into cilium assembly initiation mechanisms important for normal development.

## Methods

### Patient recruitment

The herein described cohort of 11 individuals was recruited via an international collaborative network of research sequencing laboratories, GeneMatcher Exchange platform[41], screening of public

**Fig. 8 | Interactions between NH$_2$-terminus containing the RBD and the WDR are affected by patient variants. a** Immunoprecipitation of endogenous WDR44 from controls and p.S764F patient fibroblast in the presence of 10% serum (+serum) or starved (−serum) for 24 h. Immunoblots were probed with WDR44, pAkt-substrate (pAKT WDR44), PanAKT, and β-actin antibodies. Blots are represented from two independent experiments. **b** Immunoblot analysis of Akt phosphorylation of Myc-WDR44 wild-type or G782C or S764F, and GFP-RAB11 co-immunoprecipitated under ciliating conditions (3 h and 24 h serum starvation) from 293T cells expressing indicated plasmids for 48 h. GAPDH was used to evaluate starting lysate protein levels. Blots are represented from two independent experiments. **c** The schematic represents (*top*) WDR44 domains, HA-NH$_2$-terminal domain containing RBD (1-408AA), and GST-COOH-terminal domain containing WDR (477-913 AA). Immunoblotting analysis (*bottom left*) showing co-immunoprecipitation between HA-NH$_2$-terminal domain and GST-COOH-terminal domain of wild-type or variants from 293T cells expressing indicated plasmids for 48 h. Plot (*right*) shows the relative co-immunoprecipitated levels of GST-COOH-terminal WDR44 compared to HA-NH2-terminal WDR44 proteins normalized to pre-IP levels of GST-COOH-terminal. $P = 0.0224$ (N840S), 0.0035 (L668S), 0.0021 (S764F), 0.0031 (G782C), 0.0001 (D648G), 0.0004 (D669N), 0.0008 (H839R). Mean ± s.e.m. from three independent experiments. Unpaired two-tailed *t*-test; *$P < 0.05$, **$P < 0.01$, ***$P < 0.001$. **d** Summary of WDR44 variant results. Source data are provided as a Source Data file.

databases (DECIPHER[42] https://www.deciphergenomics.org/, LOVD https://www.lovd.nl/, ClinVar https://www.ncbi.nlm.nih.gov/clinvar/), and datasets of study group consortia (TUDP) (Telethon Undiagnosed Disease Program) and the European Network of Rare Malformation Syndromes (ERN-ITHACA) or personal communication from different institutions. The study was approved by the ethics committee of Telethon Institute of Genetics and Medicine, Naples, Italy (number of protocol 81/21) and additional local ethics committees of the participating centers. We complied with all relevant ethical regulations for human patients, and written informed consent for genetic testing, publication of mutational and clinical data, and publication of patient photos and imaging was obtained from the individual's parents or legal guardian. For each affected individual, clinical data were reviewed by the clinicians (geneticists, neurologists, and pediatricians) from the participating centers, and clinical data were collected using a standardized questionnaire. Brain magnetic resonance imaging (MRI) was reviewed by a neuroradiologist expert in brain malformations (MS).

### Genetic analyses

All WDR44 variants were identified by either clinical or research ES that was performed on DNA isolated from peripheral blood of affected patients and their parents when available. Sequencing data were processed using commercial tools for the execution of the GATK Best Practices pipeline for ES variant analysis. Exon-level read counts, removal of duplicate reads, mean coverage of coding sequence regions, alignment, and variant annotation were performed using analytical pipelines that include publicly available tools and custom scripts. We looked in the index case (III:1 of family 1) at non-synonymous-exonic and splicing variants with a minor allele frequency (MAF) ≤ 0.001 in the gnomAD database (https://gnomad.broadinstitute.org/). Following their respective analysis pipelines, participating centers generated a list of candidate variants filtered against public databases and according to modes of inheritance. All variants reported in the present study were determined independently by participating centers and are classified according to the American College of Medical Genetics and Genomics (ACMG) and the Association for Molecular Pathology standards and guidelines[43,44]. Validation, the parental origin of the resulting variants, and family segregation studies were performed by Sanger sequencing (detailed conditions of the primers used, and sequencing methods are available upon request). Additional genetic testing is listed in the Supplementary material (Table S1).

### Plasmids and reagents

Wild-type cDNA of WDR44 (NM_019045.5) cloned into Gateway compatible pDONR221 (Catalog no. 12536017, ThermoFisher Scientific) vectors to generate entry clones (pDONR221-WDR44 wild-type) as described previously[11]. Myc-tagged WDR44 in the pCMV6 was acquired from Origene (Catalog no. RC205485). WDR44 variants (D648G, L668S, D669N, S764F, G782C, H839R, N840S, R733*) were generated by using primers list in Table S4 and Q5® Site-Directed Mutagenesis Kit (Catalog no. E0554S, New England Biolabs) in pDONR221 plasmid or in

the pCMV6 plasmid. pDONR-WDR44 wild-type and variants were inserted into pCS2+[11], pCS2-GFP+ (Catalog no. #1000000107, Addgene), and pDest-HA[11] using LR clonase II recombination reactions. Lentiviral plasmid pDEST686-CMV-GFP-WDR44 wild-type and variants were generated by recombining the plasmids, pDEST686, pENTR-CMV, pENTR-GFP, pDONR221-WDR44 wild-type or variants as described previously[11,45]. GFP-tagged RAB11A was acquired from Addgene (Catalog no. #56444). MigR1 vector (Catalog no. #27490, Addgene) was used for transfection of control GFP. For HA-tagged NH$_2$ fragment, GST-tagged COOH fragments, each fragment was generated by PCR using WDR44 wild-type or variants plasmid as template. Fragments were fused with tags by overlap extension PCR and were inserted into pCMV6.

Protease inhibitors (Catalog no. 539134, Roche/Millipore Sigma) and phosphatase inhibitors (Catalog no. 4906845001, Roche//Millipore Sigma) were used in immunoprecipitation and whole cell lysate analysis as per the manufacturer's recommendations. 200 nM Smoothened agonist (SAG) (Catalog no. ab142160, Abcam), 1 μM MG132 (Catalog no. ab141003, Abcam), and 10 μM chloroquine (Catalog no. C6628, Millipore Sigma) were maintained for the treatment.

### Antibodies

Commercial primary antibodies used for immunostaining and western blotting are acetylated α-tubulin (^Ac^tub) (Catalog no. T7451, Millipore Sigma), Arl13b (Catalog no. 17711-1-AP, Proteintech), CEP164 (Catalog no. sc-515403, Santa Cruz), CP110 (Catalog no. MABT1354, Millipore Sigma), Akt (pan) (Catalog no. 4691, Cell Signaling), Phospho Akt substrate (RXRXXS*/T*) 23C8D2 (Catalog no. 10001S, Cell Signaling), Phospho-AKT S473 (Catalog no. 4060S, Cell Signaling), WDR44 (also for IP) (Catalog no. A301-440A, Bethyl), β-Actin (Catalog no. 4970, Cell Signaling), β-Actin-HRP (Catalog no. A3854, Millipore Sigma), VAPA (Catalog no. 15275-1-AP, Proteintech), p27 (Catalog no. 3698, Cell Signaling), Rab11 (Catalog no. 71-5300, ThermoFisher Scientific), LC3 (Catalog no. 14600-1-AP, Proteintech), GFP (Catalog no. sc-9996, SantaCruz), GFP-HRP (Catalog no. 130-091-833, Miltenyi Biotec), Myc (Catalog no. 2276, Cell Signaling), HA (Catalog no. 3724, Cell Signaling), GST (Catalog no. 2622, Cell Signaling), GAPDH (Catalog no. sc-166574, SantaCruz), hFAB Rhodamine GAPDH antibody (Catalog no. 12004167, Bio-Rad) Hoechst (Catalog no. H3570, Molecular Probes Life Technologies).

A Cep164 antibody was generated from a protein fragment corresponding to CEP164 (CEP164N: 1–400 aa; NM_014956) that was cloned into the pGEX-4T-1. GST-CEP164N protein was expressed in BL21 CodonPlus (DE3) RIPL bacteria strain (Agilent, #230280), purified with glutathione agarose beads (Sigma, G4510). Immunization was carried out by Pocono Rabbit Farm & Laboratory (Canadensis, PA) with two chickens. Total chicken IgY was extracted from egg yolk by means of polyethylene glycol precipitation.

Secondary antibodies that were used for immunostaining and western blotting are Alexa Fluor 647 donkey anti-chicken IgY (Catalog no. 703-605-155, Jackson ImmunoReasearch Lab), Alexa Fluor 488 donkey anti-mouse (Catalog no. A-32766, ThermoFisher Scientific),

Alexa Fluor 568 donkey anti-rabbit (Catalog no. A10042, ThermoFisher Scientific), HRP-conjugated donkey anti-rabbit (Catalog no. NA934V, GE Healthcare).

## Cell lines and cell line generation

Human dermal fibroblasts (Fibroblasts) were isolated from skin biopsies as previously described[46]. p.D648G cells were from patient IV:4. Fibroblast cultures were maintained in RPMI1640 medium supplemented with 10% fetal bovine serum (FBS, Hyclone), 2% L-glutamine solution, and 1% Pen-Streptomycin at 37 °C with 5% $CO_2$.

293T (catolog no. CRL-3216, ATCC) and hTERT-RPE1 (RPE-1) (catolog no. CRL-4000, ATCC) cells were maintained in DMEM-F12 medium supplemented with 10% fetal bovine serum FBS (Hyclone) with 100 U/ml penicillin, and 100 µg/ml streptomycin at 37 °C and 5% $CO_2$ in a humidified cell culture incubator.

Rescued GFP-WDR44 wild-type fibroblasts were generated from patient fibroblasts (p.D648G and p.S764F) by using the lentivirus expression system as described previously[47]. Briefly, infected cells were selected with G418 (200 µg/ml) for a week followed by 3 days of culturing without drug treatment before using them for analysis.

WDR44 knock-out RPE-1 (RPE-1 WDR44-KO) cell line were generated using the homology-independent knock-in system[48]. Briefly, the sgRNA sequence to target human *WDR44* (5'ACTTGTTTGAGT-TAACTTCG3') was designed using Benchling life sciences R&D cloud (https://www.benchling.com) and cloned in peSpCAS9(1.1)−2 × sgRNA (Catalog no.80768, Addgene), referred to as WDR44 sgRNA plasmid. hTERT-RPE-1 cells ($1.5 \times 10^5$) were transfected with WDR44 sgRNA plasmid (1 µg) and universal donor plasmid (0.25 µg), pDonor-tBFP-NLS-Neo (Catalog no. 80767, Addgene). After 3 days cells were selected with G418 (500 µg/ml) for ~2 weeks. Cells were subsequently trypsinized as a pool and reseeded as single-cells in 96 well plates for 2 weeks to generate single-cell knock-out clones. Knock-out clones were identified by the expression of blue-fluorescence protein and analyzed for expression of endogenous WDR44 protein by immunoblotting. A knock-out clone F4 was confirmed by genotyping and sanger sequencing of genomic DNA (Fig. S6).

## Molecular Dynamics (MD) simulations and structure prediction

The WDR44 protein structure was taken from the AlphaFold Protein Structure Database[49,50] on February 14, 2022, using the UniProt identifier Q5JSH3. We modeled the core WD40 domain of WDR44 spanning residues 480-858 and 887-913. We omitted residues 859-886 because this loop region was: (1) located distal to mutation sites, (2) predicted with very low confidence by AlphaFold (pLDDT < 40) and (3) its extended structure would greatly increase water box size and therefore computational expense. The point mutation structures were built from the wild-type structure using the Visual Molecular Dynamics (VMD) mutator plugin. We omitted post-translational modifications from the WDR44 structure since we were primarily interested in assessing protein stability differences of point variants.

For MD simulations, the WDR44 core WD40 domain was solvated in a TIP3P water box with at least 15 Å water thickness around the protein. The system was then neutralized and ionized to 100 mM NaCl. Solvation and ionization were performed by VMD. The CHARMM36 protein force field was used for all protein parameters[51]. MD simulations were performed using NAMD2.13, similar to previous work[52]: 20,000 steps of solvent minimization followed by 20,000 steps of protein + solvent minimization, then 500,000 steps of equilibration with a 0.5 fs timestep. Production MD simulations were run for 500 ns with a 2 fs timestep.

Root mean square (RMSD) and root mean square fluctuation (RMSF) analyses of MD simulations were generated using Gromacs tools. From the 500 ns simulations, the last half (250 ns) was extracted and split into 50 ns portions to generate RMSD and RMSF averages. RMSD error bars represent 95% confidence intervals. Alignment of

WD40 propeller domains across proteins were used to generate the WDR44 proteomap[53].

## Stability prediction in-silico tools

To predict the impact of the variants on stability of the WD40 repeat domain the in silico stability prediction tools Genome browser GRCh38/hg38 (https://genome.ucsc.edu/), PhyloP100way[54], GERP++[55], dN/dS (MetaDome)[56], PolyPhen2[57], PROVEAN[58], MUpro[59], SIFT[60], MutationTaster[61], CADD[62], and REVEL[63] were used to analyze seven missense variants identified in patients.

## Immunofluorescence

Immunofluorescence studies were performed on 4% PFA fixed and immunostained cells as described previously[13]. Imaging was done by using a 40 × 1.4 numerical aperture (NA) (for ciliation) or 63 × 1.3 NA (for CP110 level) objective in Zeiss Axio Scan Z1 inverted epifluorescence microscope equipped with a CoolSNAP HQ2 camera or Zeiss LSM710 confocal microscope. Slidebook software (Intelligent Imaging Innovations) or ZEN software (Zeiss) was used for image acquisition and analysis. Ciliation was quantified from >150 cells from three independent experiments (unless otherwise indicated) stained with Arl13b (cilia marker) and CEP164 (MC marker). CP110 levels at the MC were quantified from >100 cells as previously described[47].

## Immunoblotting and immunoprecipitation

Fibroblasts, 293T, and RPE-1 cell lines were lysed in RIPA buffer with protease and phosphatase cocktail inhibitors separated SDS-PAGE and analyzed by immunoblotting. Enhanced chemiluminescence (ECL) was used to develop immunoblots, and protein levels were quantified using ImageJ.

Immunoprecipitation of endogenous WDR44 from fibroblasts or GFP-WDR44 wild-type and variants expressing 293T cells were done as described[11]. Briefly, ~70−80% confluent fibroblasts were maintained with serum or starved for 24 h. Fibroblasts were washed with PBS, lysed in low-salt triton buffer (75 mM NaCl, 30 mM Tris HCl pH 8.0, 10% Glycerol, 1% Triton X-100) supplemented with 5 mM $MgCl_2$, and protease and phosphatase inhibitors, followed by immunoprecipitation with Protein A beads conjugated with WDR44 antibody (Catalog no. A301-440A, Bethyl) for 3 h. Protein A beads were washed with low-salt triton buffer, and samples were resolved by SDS-PAGE and analyzed by immunoblotting with appropriate antibodies.

293T cells were transfected with indicated plasmids using X-tremeGENE™ 9 (Catalog no. 6365779001, Millipore Sigma) or Lipofectamine 2000 (Catalog no. 11668019, Invitrogen). After 48 h of transfection, cells were washed with PBS and lysed in low-salt triton buffer supplemented with 5 mM $MgCl_2$, and protease and phosphatase inhibitors. GFP-fused proteins were immunoprecipitated using GFP-Trap affinity beads (Chromotek, gta-20). For Myc or HA-tagged antibody immunoprecipitations cells were lysed in 150 mM NaCl, 20 mM Tris HCl pH 8.0, 10% Glycerol, 1% Triton X-100 and incubated with Protein G Sepharose® beads (Catalog no. 17061801, GE Healthcare). Beads were washed with low-salt triton buffer and samples were resolved by SDS-PAGE and analyzed by immunoblotting with appropriate antibodies. Akt-dependent WDR44 phosphorylation levels were determined by using an Akt substrate antibody. Total Akt was analyzed in the cell lysates by immunoblotting with the Akt (pan) antibody.

## Zebrafish analysis

Zebrafish AB and Tg (*wt1b:EGFP*, a generous gift from Christoph Englert, PhD, Fritz Lipmann Institute and Michael Tsang, PhD, Pittsburg University), were maintained in accordance with the protocols (ASP#20-416) approved by the Animal Care and Use Committee of the National Cancer Institute at Frederick and AAALAC guidelines. RNA injections and analysis of 2−6 days old embryos were performed as described previously[47]. Briefly, full-length sequences of human *WDR44*

wild-type and variants were subcloned into pCS2+ and pCS2-GFP+ vectors (Addgene, Kit#1000000107), and messenger RNAs were transcribed using the mMESSAGE mMACHINE SP6 kit (Ambion) according to manufacturer's instructions. Embryos were injected with 100 pg/nl of capped mRNAs at the one-cell stage. For morpholino knockdown and rescue experiments, 250 μM of wdr44 morpholino 1 (MO1) (5′GCTGCATACAGAGGCCGCCGCCACT-3′)[11] or morpholino 2 (MO2) (5′GCTACTCTCTGGACGAAGCAGACTA −3′) custom-synthesized by Genetools was injected in combination with WDR44 variants or wild-type RNAs. The efficiency of morpholinos was confirmed by coinjecting mRNA transcribed from pCS2-MO1/2$^{Target}$-GFP. The pCS2-MO1/2$^{Target}$-GFP template was engineered with a pCS2-GFP+ vector that contains the zebrafish target (TAGTCTGCTTCGTCCAGAGAGTAGCAGTGGCGGCGG CCTCTGTATGCAGCCGGGCTCAATGG) wdr44 5′UTR sequence downstream of the GFP sequence. For the CRISPR methodology, Alt-R CRISPR-Cas9 gRNA were designed using the IDT custom Design Tool (https://www.idtdna.com/site/order/designtool/index/CRISPR_CUSTOM). Alt-R CRISPR-Cas9 sgRNA (Dr.Cas9.WDR44.1.AC, 431151565) and Alt-R S.p. Cas9 Nuclease V3 (IDT, 1081058) protein were prepared following the manufacturer's instructions. The gRNA was combined with 0.5 μg/μl of Cas9 protein to form the RNP complex using a 1:1 concentration ratio and incubated at 37 °C for 10 min. Microinjection was performed by injecting ~3 nl of RNP complex into the yolk of one-cell-stage embryos.

For IFM studies, embryos were fixed with a solution of 4% PFA + 0.25% TX100 and stained with Phalloidin conjugated with Alexa 568 (Catalog no A12380, Molecular Probes Life Technologies), anti-Acetylated tubulin (Catalog no. T6793, Sigma), and Hoechst (Catalog no. H3570, Molecular Probes Life Technologies). Embryos were imaged by spinning disk confocal (SDC) microscopy using a 40 × 1.4 NA oil objective on a Marianas SDC (Intelligent Imaging Innovations). Ciliated organ assessment at 3 dpf was performed as follows: organs with absent or strongly reduced cilia numbers were quantified and this data was used to calculate the percentage of normal versus total number of organs observed. HCR analysis was performed as previously described refs. [64] using HCR probe sets targeting wdr44 (NM_001100038.2) and elavl3 (NM_131449) designed and provided by Molecular Instruments. Protein detection and in situ HCR were performed following the manufacturer's instructions. For dot blot analysis, embryos were homogenized in RIPA buffer supplemented with protease and phosphatase inhibitors. Lysates were centrifuged at 10,000 g for 10 min. Cleared supernatant was collected and 3μl was spotted on nitrocellulose membrane with pore size 0.2μm (Bio-Rad) and air-dried for 30 min. The membrane was washed with Tris-buffered saline (TBST) and blocked in 5% non-fat dry milk prepared in TBST buffer for 1 h at room temperature. The membrane was incubated with the WDR44 antibody overnight at 4ºC, washed with TBST, followed by incubation with the secondary HRP-conjugated anti-rabbit antibody and incubated with ECL and exposed to film. Total protein levels applied to the membrane was visualized by staining the immunoblot with Ponceau S Solution (Catalog no. P7170, Millipore Sigma). Western blotting analysis was performed as previously described[47]. Briefly, embryos were homogenized in lysis buffer (20 mM Tris at pH 8, 137 mM NaCl, 10% glycerol, 1% Triton X-100) with protease inhibitor cocktail (Roche) and centrifuged at 13,000 r.p.m for 10 min. Supernatants were collected and sample buffer was added prior to boiling.

### RT-PCR analysis
RNAs were isolated from control and patient fibroblasts using Trizol reagent according to the manufacturer's directions. WDR44 (Hs00219373_m1), PTCH1 (Hs00181117_m1), GLI1 (Hs01110766_m1), and GAPDH (Hs99999905_m1) expressions in total RNA of control and patient fibroblasts were determined using Taqman probes purchased from Applied Biosystems and the one-step qScriptTM XLT kit from Quanta Biosciences according to manufacturer's instructions. Quantitative RT-PCR was performed using the Applied Biosystems 7500 Fast Real-Time PCR system. For p.R733* and male control expression analysis mRNA isolation was performed using RNeasy kit (Qiagen) and the Luna® Universal One-Step RT-qPCR Kit (New England Biolabs) was used according to the manufacturer's instructions with quantitative RT-PCR performed with a CFX96 Bio-Rad thermocycler.

### Statistical analysis
GraphPad Prism 9 was used for the statistical data analysis and means ± s.e.m or s.d. are specified in the figure legends. Unpaired two-tailed Student's t-test was applied for comparisons with control fibroblasts or siControl or as indicated in figure legends. The number of zebrafish embryo or ciliated organs counted is indicated by n. Cell numbers counted for cilia determination are indicated in methods or figure legends. For randomization of zebrafish experiments, each injected group was assigned a random number, which was not revealed to the investigator until the final outcome assessment.

### Reporting summary
Further information on research design is available in the Nature Portfolio Reporting Summary linked to this article.

## Data availability
All data that support the findings in this paper are available within the article and its supplementary information files. Source data are provided in the paper. Source data are provided with this paper.

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

## Acknowledgements

We are grateful to the patients and families who participated in this investigation. This research was supported by the Intramural Research Program of the National Cancer Institute contract #HHSN26120080001E to C.J.W., the Intramural OPBG Research Program supported by Fondazione Bambino Gesù (Vite Coraggiose) to M.T., Italian Ministry of Health (RCR-2022-23682289, PNRR-MR1-2022-12376811 and Ricerca Corrente 2019 to M.T.), Italian Ministry of Research (FOE_2020 to M.T.), the National Research Foundation of Korea (NRF-2022M3A9I2017587) to S.G.P. and the Kun-hee Lee Seoul National University Hospital Child Cancer & Rare Disease Project, Republic of Korea (grant number: 22B-001-0500) to S.G.P. A.A. acknowledge the TUDP for performing exome sequencing of family 1 with the Telethon project GSP15001. The content of this publication does not necessarily reflect the views or policies of the Department of Health and Human Services, nor does mention of trade names, commercial products, or organizations imply endorsement by the U.S. Government. This study makes use of data generated by the DECIPHER community. A full list of centers who contributed to the generation of the data is available from https://deciphergenomics.org/about/stats and via email from contact@deciphergenomics.org. Funding for the DECIPHER project was provided by Wellcome [grant number WT223718/Z/21/Z].

## Author contributions

A.A., S.S., T.Y., C.I., H.J.C, S.G.P., and C.J.W. designed the study and analyzed data. S.S., T.Y., C.I., D.B., P.Y., J.S., and K.R.B. performed the experiments. S.S., H.Z., H.S.L., and H.J.C. generated reagents. A.A., S.Y.K., T.Y., M.C., J.H.C., A.D., A.G., C.F., L.S.R., Y.R., S.R.L., and R.L. provided clinical data. D.B., K.J.W., P.Y., M.N., M.T., S.P., and G.C. performed structure modeling. M.S. reviewed neuroimaging. A.A. and C.J.W. wrote the manuscript. S.S., T.Y., C.I., S.Y.K., and S.G.P. contributed to the writing of the manuscript. The following authors contributed to patients recruitment and assessment; C.F., M.P., M.C.D., S.B., V.Z., A.S., L.P., H.W.M., N.L.C., G.H., E.C., V.C., J.D., and A.L. The following authors contributed to genetic studies; M.S., S.B., J.W., J.A.R., T.M.S., D.S., M.S.A., A.T., V.N., R.P., V.S., F.Z., M.O., J.D., and M.J.L. All authors critically reviewed the manuscript.

## Competing interests

J.A.R. declared that the Department of Molecular and Human Genetics at Baylor College of Medicine receives revenue from clinical genetic testing completed at Baylor Genetics Laboratories. The remaining authors declare no competing interest.

## Additional information

**Andrea Accogli**[1,2,38], **Saurabh Shakya** ®[3,38], **Taewoo Yang**[4,38], **Christine Insinna**[3], **Soo Yeon Kim**[5], **David Bell**[6], **Kirill R. Butov**[7,8], **Mariasavina Severino**[9], **Marcello Niceta** ®[10], **Marcello Scala** ®[11,12], **Hyun Sik Lee** ®[13], **Taekyeong Yoo**[14], **Jimmy Stauffer**[3], **Huijie Zhao** ®[3], **Chiara Fiorillo**[11,15], **Marina Pedemonte**[12], **Maria C. Diana**[12], **Simona Baldassari**[16], **Viktoria Zakharova** ®[17], **Anna Shcherbina**[7], **Yulia Rodina**[7], **Christina Fagerberg** ®[18], **Laura Sønderberg Roos**[19], **Jolanta Wierzba**[20], **Artur Dobosz**[21], **Amanda Gerard**[22,23], **Lorraine Potocki**[22,23], **Jill A. Rosenfeld** ®[23,24], **Seema R. Lalani**[22,23], **Tiana M. Scott**[25], **Daryl Scott** ®[24], **Mahshid S. Azamian** ®[24], **Raymond Louie**[26], **Hannah W. Moore**[26], **Neena L. Champaigne**[26], **Grace Hollingsworth**[26], **Annalaura Torella**[27,28], **Vincenzo Nigro**[27,28], **Rafal Ploski**[29], **Vincenzo Salpietro** ®[30,31], **Federico Zara** ®[11,16], **Simone Pizzi**[10], **Giovanni Chillemi** ®[32], **Marzia Ognibene** ®[16], **Erin Cooney**[33], **Jenny Do**[33], **Anders Linnemann**[34], **Martin J. Larsen** ®[18,35], **Suzanne Specht**[3], **Kylie J. Walters** ®[36], **Hee-Jung Choi** ®[13],

Murim Choi ⓘ [14], Marco Tartaglia ⓘ [10], Phillippe Youkharibache[37], Jong-Hee Chae[5], Valeria Capra[15], Sung-Gyoo Park ⓘ [4] ✉ & Christopher J. Westlake ⓘ [3] ✉

[1]Division of Medical Genetics, Department of Specialized Medicine, McGill University Health Centre (MUHC), Montreal, QC, Canada. [2]Department of Human Genetics, McGill University, Montreal, QC, Canada. [3]Laboratory of Cell and Developmental Signaling, Center for Cancer Research, National Cancer Institute, National Institutes of Health, Frederick, MD, USA. [4]Institute of Pharmaceutical Sciences, College of Pharmacy, Seoul National University, 08826 Seoul, Republic of Korea. [5]Department of Genomic Medicine, Seoul National University Hospital, 03080 Seoul, Republic of Korea. [6]Advanced Biomedical Computational Science, Frederick National Laboratory for Cancer Research, Frederick, MD, USA. [7]Department of Immunology, Dmitry Rogachev National Medical Research Center of Pediatric Hematology, Oncology and Immunology, Moscow 117997, Russia. [8]Department of Molecular Biology and Medical Biotechnology, Pirogov Russian National Research Medical University, Moscow 117997, Russia. [9]Neuroradiology Unit, IRCCS Istituto Giannina Gaslini, Genoa, Italy. [10]Molecular Genetics and Functional Genomics, Ospedale Pediatrico Bambino Gesù, IRCCS, 00146 Rome, Italy. [11]Department of Neurosciences, Rehabilitation, Ophthalmology, Genetics, Maternal and Child Health, Università Degli Studi di Genova, Genoa, Italy. [12]Pediatric Neurology and Muscular Diseases Unit, IRCCS Istituto Giannina Gaslini, Genoa, Italy. [13]School of Biological Sciences, Seoul National University, 08826 Seoul, Republic of Korea. [14]Department of Biomedical Sciences, Seoul National University College of Medicine, 03080 Seoul, Republic of Korea. [15]Child Neuropsychiatry, IRCCS Istituto G.Gaslini, DINOGMI University of Genova, Largo Gaslini 5, Genoa, Italy. [16]Unit of Medical Genetics, IRCCS Istituto Giannina Gaslini, 16147 Genoa, Italy. [17]National Medical Research Center for Endocrinology, Clinical data analysis department, Moscow, Russian Federation, Russia. [18]Department of Clinical Genetics, Odense University Hospital, Odense, Denmark. [19]Department of Clinical Genetics, Rigshospitalet, Copenhagen University Hospital, København, Denmark. [20]Department of Pediatrics and Internal Medicine Nursing, Department of Rare Disorders, Medical University of Gdansk, Gdansk, Poland. [21]Department of Medical Genetics, Faculty of Medicine, Jagiellonian University Medical College, 30-663 Krakow, Poland. [22]Texas Children's Hospital, Houston, TX, USA. [23]Department of Molecular and Human Genetics, Baylor College of Medicine, Houston, TX, USA. [24]Baylor Genetics Laboratories, Houston, TX, USA. [25]Division of Microbiology and Immunology, Department of Pathology, University of Utah School of Medicine, Salt Lake City, UT 84112, USA. [26]Greenwood Genetic Center, Greenwood, SC, USA. [27]Telethon Institute of Genetics and Medicine (TIGEM), Naples, Italy. [28]Department of Precision Medicine, University of Campania "Luigi Vanvitelli", Naples, Italy. [29]Department of Medical Genetics, Medical University of Warsaw, Pawińskiego 3C, 02-106 Warsaw, Poland. [30]Department of Neuromuscular Disorders, Queen Square Institute of Neurology, University. College London, London WC1N 3BG, UK. [31]Department of Biotechnological and Applied Clinical Sciences, University of L'Aquila, 67100 L'Aquila, Italy. [32]Department for Innovation in Biological, Agro-food and Forest systems, DIBAF, University of Tuscia, Via S. Camillo de Lellis s.n.c, 01100 Viterbo, Italy. [33]Division of Medical Genetics and Metabolism, Department of Pediatrics, University of Texas Medical Branch, Galveston, TX, USA. [34]Hans Christian Andersen Children's Hospital, Odense University Hospital, Odense, Denmark. [35]Clinical Genome Center, Department of Clinical Research, University of Southern Denmark, Odense, Denmark. [36]Center for Structural Biology, Center for Cancer Research, National Cancer Institute, National Institutes of Health, Frederick, MD, USA. [37]Cancer Science Data Lab, Center for Cancer Research, National Cancer Institute, National Institutes of Health, Bethesda, MD, USA. [38]These authors contributed equally: Andrea Accogli, Saurabh Shakya, Taewoo Yang. ✉e-mail: riceo2@snu.ac.kr; chris.westlake@nih.gov

