## [Peer Review File · Nature Communications]

Variants in the WDR44 WD40-repeat domain cause a spectrum of ciliopathy by impairing ciliogenesis initiationREVIEWER COMMENTS

Reviewer #1 (Remarks to the Author):

The authors identified pathogenic variants in WDR44 as an X-linked cause of intellectual disability, hypotonia and various other features. They show that the variants tend to result in impairment of ciliogenesis initiation and signalling. They also show that the variants tend to result in a gain of function of RAB11 binding, which contributes to dysregulation of ciliogenesis initiation. This is the first report that associates WDR44 variants with a clinical phenotype in humans. The authors provide adequate support for their clinical findings. The collaborative effort that led to this paper is impressive and it is not that usual that such a large series of patients is reported in a cilia gene discovery these days. There is some room to solidify the genetic findings further. In families in which the variant was maternally inherited testing of the maternal parents may benefit variant interpretation in small families, particularly for VUS. Some variants are more convincing than others in terms of in silico predictions and functional studies; p.N840S is a variant that I would consider as a variant of uncertain significance at best. Generally, additional experimentation to verify endogenous localization/expression patterns with WDR44/Wdr44 and RAB11 antibodies could further solidify functional data. Overall, this report is certainly of interest to the readers of Nature Communications after major/minor comments have been addressed. The report is well written – it was a pleasure to read. Thank you for submitting robust work and congratulations on your exciting findings.

Major:

- If maternal grand parents are available for this study, it would further support the findings when only grandmothers are carriers in the assessed families for each of the variants (except family 4 as this family already has a supportive and strong pedigree, and except families 2 and 7 as de novo status was reported). Are the variants in families 2 and 7 “confirmed de novo” and maternity/paternity verified during WES data analysis?
- The authors wait until the discussion of this report with saying that the p.N840S variant may be mild or possibly benign. Can the authors please state that this is a variant of uncertain significance earlier in the report (e.g. in the last section of ‘Identification of WDR44 variants’ on page 8)? The authors should also add to this section that while the conservation is high for c.2519A>G, the REVEL score is in the tolerated range. The score for 2003T>C (p.L668S) is in the borderline damaging/uncertain range at 0.6299; it would be worthwhile to also mention that in the main text of the report. Please, feel free to also add that REVEL scores for the other variants are all in the damaging range. Please note that REVEL is written as RELEV in Table S2.
- The nonsense variant c.2197C>T is not located in the last or penultimate exon of WDR44. It is expected to lead to nonsense mediated decay and predicted to lead to an absent or an abnormal protein. The authors say the following about this variant ‘ The nonsense variant in exon 16 introduces a stop that is predicted to truncate the protein at the residue 733 in WD5’ on page 8. I think they need to explain on page 8 that the nonsense variant could lead to both NMD and/or an abnormal protein, which may or may not be truncation at codon 733. Rightfully, the authors do acknowledge the possibility of escape of NMD in their discussion. They say that this phenomenon is ‘not uncommon’ and refer to only a single paper where escape of NMD was reported as an example. Do the authors know what percentage of nonsense variants located in exons that are not the last or penultimate exon generally escape NMD? What is ‘not uncommon’? I think of escape of NMD as something that is rare, but I might be wrong... If possible, it would be great if the authors could explore RNA/protein expression in fibroblasts of the patient with the c.2197C>T nonsense variant to solidify their hypothesis that the c.2197C>T variant is associated with an expressed abnormal WDR44 protein that leads to a gain-of-function effect.
- Figure 7c. If the authors have access to patient-derived cells (other than N840S) and WDR44/RAB11 antibodies, what do endogenous localizations look like in patient-fibroblasts versus controls? Are WDR44 and RAB11 expressed in human fibroblasts? It would be very compelling if Figure 7c findings are verified with endogenous data.
- The authors should briefly specify the most common clinical features in the main abstract of the paper and at least mention ID, hypotonia, characteristic facial features as all patients had at least these features along with variable other characteristics.
- Please do not overstate your conclusions. I like that data is summarized per section, but the detected variants do not all have similar effects and therefore it is important to phrase your

conclusions more carefully. Some examples: Page 13: Thus, these results indicate protein abundance of WDR44 variants is also affected in zebrafish >> Thus, these results indicate protein abundance of various WDR44 variants is also affected in zebrafish. Page 13: Together, these results recapitulate the pathogenicity of the human WDR44 variants associated with ciliopathy phenotypes >> Together, the majority of WDR44 variants recapitulate the WDR44-associated human features in zebrafish. Page 14: ...suggesting that WDR44 variants exert GOF in repressing ciliogenesis >> ...suggesting that various WDR44 variants exert GOF in repressing ciliogenesis.

Minor:

- The use of the word 'mutations' (e.g. page 2, 6, probably others) is out of date and should be replaced with 'variants' or 'pathogenic variants' (whatever is relevant at each section) throughout the manuscript.
- Page 2: .. encoded by X-linked gene > ...encoded by the X-chromosomal gene WDR44 (OMIM *301070).
- Page 6: ... Ciliopathy-associated missense variants show enhanced proteasomal degradation... >> WDR44 containing ciliopathy-associated missense variants is subject to enhanced proteasomal degradation
- The resolution of all pedigrees in Figure S1 is poor and should be replaced with images with improved resolution. The text is difficult to read.
- Figure 1 and S1, the images of the patient with the p.D648G variant seem squished?!
- In protein nomenclature there should not be any spaces between the p. and the remainder of a variant, e.g. p. S764F should be p.S764F. Please check nomenclature throughout the manuscript for all variants in the main manuscript, legends and supplemental documentation.
- One could add in the manuscript that the Z-score of the WDR44 gene in gnomAD is 2.95, which provides some support for intolerance of missense variation in this gene. Null variants in this gene are certainly selected against.
- Page 7: The authors say that they assessed WES trio in their first patient (p.S764F) for LP/P variants in known disease genes, and that they subsequently focused on ciliary genes. Did the authors also use alternate prioritizations? For example, all genes known to be expressed in brain development? Were there any other candidate genes found?
- It is somewhat surprising to me that unilateral multicystic kidney findings were noted in patient with p.S764F because as far as I am aware renal cysts tend to be bilateral in ciliopathies. Are there other reports of cilia dysfunction and unilateral renal cyst findings? How do the authors explain this? Do they suspect the other kidney may also develop cysts in time? Was there anything remarkable in the imaging of the other kidney?
- S4f. Is there a Wdr44 antibody available? If so, can the authors show where/when endogenous Wdr44 expresses in zebrafish?
- Page 14: ... In contrast, morphants expressing WDR44 variants, except for D648G and L668S, caused defective ciliation in the neuromasts, olfactory placodes and pronephric ducts. >>> In contrast, morphants expressing WDR44 variants, except for D648G, L668S and N840S, caused defective ciliation in the neuromasts, olfactory placodes and pronephric ducts.
- Figure 7A clearly shows increased RAB11A binding in the majority of mutants compared to wild type. The authors state that the same can be seen in Figure 7B with endogenous RAB11A, but RAB11A binding seems somewhat reduced for G782C (maybe because of overall low GFP-WDR44 expression in WCL?). Can you please compile a similar graph as shown in Figure 7A: it may help to interpret Figure 7B? Similarly, binding with VAPA seems to be lowish in various mutants? Is that just artificial variation of the assay or is there is an additional variant effect?
- The authors state in the discussion that cerebral and cerebellar white matter changes may be due to ciliary defects. This is a possibility, but it is also possible that this phenotype is due to disruption of an extra-ciliary function of WDR44. The authors say that WDR44's function is not limited to cilia earlier in the manuscript and in the final conclusion as well, but I think it is worthwhile to also mention this in the white matter discussion because that phenotype does seem rather unique.

Reviewer #2 (Remarks to the Author):

The manuscript by Acogli et al describes a new gene-disease association with mutations in WDR44 in 10 patients presenting with complex neurodevelopmental and multisystemic disease. The authors demonstrate that the variants, which are mostly missense variants, lead to decreased expression levels of the mutant WDR44 protein through proteasomal degradation, but suggest that the remaining WDR44 variant protein acts through a gain-of-function mechanism with increased WDR44-RAB11 binding, which in turn affects ciliogenesis by decreasing the ability of RAB11 to release the CP110 capping required in the first steps of ciliogenesis. Overall, this represents an interesting story; the manuscript is overall well written and structured. However, some parts of the results are not as compelling as they could be: while the association between the described WDR44 variants and human disease appears to be strong, including the effect of the variants on binding to RAB11, the actual direct link to cilia appears less convincing. The authors claim that this is a ciliopathy based on the patients' clinical phenotype, on the zebrafish data and on the ciliation experiments in cell lines. I would argue however, that (1) the phenotypic constellation of the patients described is not really typical for bona fide ciliopathies (see details below); (2) the zebrafish data is moderately convincing due in part to some technical problems (see details below); (3) the cell culture experiments are mostly performed with overexpression experiments in lines that still contain endogenous protein (which is not addressed at all) and fibroblasts were available only from 3 patients; moreover, the data on ciliation are somewhat confusing/show different results between the patients and between fibroblasts and overexpression experiments in various cells types, which weaken the conclusion that the WDR44 variants cause the phenotype through a direct effect on ciliogenesis and that the patients' disorder is a "first order" ciliopathy. Detailed main points:

(1) patient phenotypes as an indication for a ciliopathy:

the patients do present with pleiotropic disease, which does affect some of the organ systems that are also commonly affected in ciliopathies, but the way these organs are affected is not really typical or specific for the classical ciliopathies (for example, while there is musculo-skeletal involvement in these patients, this is not a patterning defect with polydactyly as typical for ciliopathies or a skeletal dysplasia as in the short rib polydactylies or other skeletal ciliopathies; brachydactyly, syndactyly or scoliosis are all found in many other non-cilia-related disorders, probably even more often than in ciliopathies; only one patient has cysts in one kidney and one patient has a nephritis, so renal involvement is also not prominent in this cohort (other renal symptoms such as nephrolithiasis or renal hypoplasia are not cilia-related phenotypes); the CNS anomalies mostly appear to affect white matter, which is not typical for ciliopathies, and in a few cases the corpus callosum, but this is also seen in many non cilia-related syndromes). So all in all, calling the patients' phenotypes "ciliopathy-related" per se and using this as an argument to support the link between these variants and a direct link to ciliary defects seems a bit of an over-interpretation (they might be compatible with ciliary dysfunction but cannot be seen as typical for this). Other syndromic disorders have been shown to have an indirect effect on cilia and ciliogenesis and can be described as "second-order ciliopathies". This possibility should at least be mentioned. Depending on the extent to which the authors can address the following comments, I think that the authors may have to be more conservative in their overall interpretations.

(2) zebrafish experiments:

- One problem with the ZF experiments is that morpholinos were used to knock-down endogenous wdr44; it remains unclear why this approach was chosen (the rationale underlying the statement "To closer mimic the hemizygous state of WDR44 patients we used morpholinos (MOs) to reduce expression of the endogenous Wdr44 protein" is hard to understand. For an X-linked gene the amount of protein is the same in a hemizygous male as for an autosomal gene in a homozygous individual as far as I know. So I am not sure what the advantage would be to titrate the amount of gene knockdown in this situation (plus, an actual titration is not described in this work). Moreover, the community now usually agrees that morpholinos should only be used with proper controls which include comparison with a mutant (see recommendations paper doi: 10.1371/journal.pgen.1007000). Particularly in this situation, where the authors then overexpress the variants, it would have been much cleaner to do so in a knock-out background. I can see the point that the morphants have no phenotype (see point below) so there is not much to compare to a mutant, but maybe this would be a good reason to see what happens when the gene is knocked-out. The authors also provide insufficient information on the morpholinos used: are these ATG morpholinos or splice morpholinos? In the latter case, a splicing study would have been nicer to

show the effect of the morpholino (rather than the effect on GFP expression of a co-injected construct).

- It is also surprising that the morphants display no phenotype at all (how closely were these analyzed? Is there no effect on cilia at all and how does this fit with the knock-out data in cell lines which exhibit higher ciliation rates?)

- Moreover, the WB shows that none of the mutant forms of the protein is detected in zebrafish lysates except for N840S (figure S4e), which questions how the effect can be caused by a mutant protein which is not detectable (the possibility that the variant affects the antibody epitope is not supported by the fact that the same mutations are detected in human fibroblasts). In figure 4i it is also rather hard to understand how this was quantified and how the GFP results fit with the WB data shown in figure S4e?

- Finally, and most importantly, the phenotypes shown after overexpression of the mutant Wdr44 forms can hardly be considered "typical ciliopathy phenotypes": the kidney cysts are not visible on several of the images in figure 4h (why was this looked at at 6dpf rather than 3dpf where the MO and the mRNA are likely still active while at 6dpf their effect could be gone?), and the remaining phenotypes look in fact just as similar to non-specific toxicity effects of mRNA/MO injection (heart edema, small heads, crooked fish). A recent paper with comprehensive comparison of multiple ZF ciliopathy mutants has not found any of these features (doi: 10.1242/dmm.049568). I acknowledge the fact that the WT mRNA did not appear to cause any phenotypes, which would argue that the mutant forms have an additional effect, but unfortunately, I'm not sure this suffices to claim specificity of a gain-of-function effect of the mutant protein (which is not detectable by WB) on cilia-related phenotypes. The cilia-defects shown in figure 5 are also not so convincing: in several images, the actual organ analyzed (neuromast, olfactory placode) appears very underdeveloped, likely in line with the fish being rather sick, and so it's difficult to interpret an effect on cilia in a malformed organ. It is also unclear how the quantifications were performed (yes/no cilia anomalies per embryo?)

Ideally, the authors should express less wdr44 mRNA in a stable mutant knock-out background to make these results more compelling. This does represent a substantial additional amount of work, but could substantially strengthen the link to cilia. I would recommend at the very least to characterize the morphant phenotype better and to compare it to F0 crispants to have an alternative method for interfering with the endogenous gene. Most importantly, I would recommend decreasing the dose of injected mRNA so as to have embryos that look less sick and analyze the cilia in these in a more stringent manner (actual quantification of ciliation in selected organs).

(3) Cell experiments:

- The most compelling data concerning the link between novel gene variants and ciliation often comes from patient fibroblast data, since this represents the closest situation to the disease state without additional interference; unfortunately these are generally hard to get and indeed, in this study, lines were available only from 3 patients. What is a little disturbing, is that the ciliation results differ between these three patients (while the significance of N840S is unclear as also discussed by the authors, and this line may be considered separately from the others throughout the paper, the other two lines should behave more similarly). Some degree of difference in severity is of course acceptable, but it is rather disturbing that only S764F can be rescued with respect to ciliation and not D648G. This substantially weakens the claim that WDR44 variants cause a phenotype through a defect in ciliation. Then the authors drop the D648G line in the following experiments. So in the end, the only really strong and consistent data supporting an effect of WDR44 variants on ciliation is on one variant (S764F).

- The authors use a variety of assays in a variety of cell lines which give somewhat different results that make things rather confusing: (1) 2 patient fibroblast lines have no cilia without serum, but only one line has normal ciliation with serum; the third line (N840S) has no effect on ciliation but has shorter cilia; (2) expression of myc-tagged WDR44 variants in 293T cells (where endogenous protein is not analyzed/interfered with) shows no effect on ciliation rate but results in shorter cilia for all variants. (3) The authors then go on and make RPE1 KO lines for WDR44 and analyze CP110 capping after expression of selected WDR44 variants (why only some?) in the RPE1 KO lines (and in only one patient fibroblast line). (4) Then the authors go back to analyzing RAB11 localization in 293T cells. Why they go back to 293T cells is unclear. In fact, now that the authors have made a KO line lacking endogenous WDR44, the results could be much more compelling if all the variants were expressed and analyzed in this null background: ciliation for all variants could be assessed in this setting, in parallel with the CP110 capping, the HH assay and the RAB11 assay.

Consistent results with such a more unified set of experiments would substantially strengthen the claim that these WDR44 variants cause disease through an effect on ciliogenesis. Similarly, it should be possible to look at CP110 and RAB11 in the patient fibroblast lines that are available (all 3), in addition to the ciliation analysis.

Comments per figure:

Figure 1:

Some of the patient figures appear to be distorted (panel 2 and probably 1 of figure 1a and panel b of figure S1). The image resolution was too low to be able to increase the size of the MRI images sufficiently to actually see things well enough.

Figure 2:

2c: why are the stats not indicated for the comparison between WT and the truncating variant? If this is not significant, it might be better not to mention in the text that there is a trend for an increased expression?

2e: The control with MG (panel e) also appears to show a much stronger band, which is not reflected in the quantification graph?

2c compared to 2e: in 2c, several variants show very decreased expression but in 2c with chloroquine (almost) no difference?

Figure 3:

3b: why are ciliation levels not restored in D648G but only in S764F? If dominant negative / gain-of-function effect, than overexpressing the WT form should not rescue either, but if loss-of-function effect, than WT should rescue both?

3d-e: no data on D648G?

Figure 4:

Quantifications are yes/no per embryo to reach the % shown?

The term "scoliosis" is not really appropriate for a zebrafish larva that has no skeleton at this point. Body curvature would be a more descriptive and conservative term.

4h: cysts are not visible in several panels (when enlarging, the resolution is insufficient)

4i: unclear how the +, ++ and +++ were determined; not obvious from the images

Figure 5:

Unclear how the quantifications were decided (as above)

5a shows no phalloidin

Figure 6:

6a: why is there no data shown on the fibroblast line of patient D648G?

Figure 7:

7b: I am not sure that the difference between RAB11 and VAP is very compelling; the text in the corresponding paragraph is also written in a somewhat confusing manner.

Discussion:

In the discussion the emphasis on the "ciliopathy phenotypes" appears too strong (see comments above). Some parts of the discussion also are bitt too much of a repetition of results with detailed re-discussion of each variant and reference to figures again; maybe this should be summarized a little more.

Accogli et al. Responses to reviewer comments:

Responses to Reviewer #1:

Major:

- If maternal grand parents are available for this study, it would further support the findings when only grandmothers are carriers in the assessed families for each of the variants (except family 4 as this family already has a supportive and strong pedigree, and except families 2 and 7 as de novo status was reported). Are the variants in families 2 and 7 “confirmed de novo” and maternity/paternity verified during WES data analysis?

Response: We thank the reviewer for this suggestion. We were able to collect additional information from patient family members that further support our findings. We extended segregation analysis in male relatives (Figure S1a of the revised manuscript) in families 1, 7 (former family 6), 8 (former family 7) and 9 (former family 8). Target variant testing in 4 first degree maternal uncles of family 1 and proband’s brother, and maternal grandfather, and 2 maternal uncles of family 8 (former family 7) showed that these healthy male individuals were wild type for the p.S764F and p.H839R variants, respectively.

New segregation analysis in family 9 showed that the p.N840S variant was also present in a patient’s first degree cousin (III:4). Due to limited clinical information we have not included this patient in this manuscript. Still this patient was described as having mild speech delay, febrile seizure and ADHD consistent with male relative (III:2). Our previous thoughts are in line with the extended segregation suggesting possible variable expressivity and minor or no consequences for protein function. We agree with the reviewer that the p.N840S variant has uncertain significance and we have further commented on this in the revised manuscript.

We have also performed segregation analysis in family 7 in which mother resulted mosaic ($\approx 20\%$) for the variant p.D669N, suggesting this was a post-zygotic event.

Regarding family 3, the maternal grandfather was deceased and mother does not have any brothers to extend the segregation.

Regarding families 2 and 5 in which p.G782C and p.L668S were *de novo* the following analysis have been performed:

- Family 2: The maternity/paternity of family 2 was tested by analyzing all high quality autosomal SNVs from the proband for Mendelian inheritance. Indeed, majority (99.88%) of the variants were inherited to the proband in expected patterns (Table S4), demonstrating that they are the true biological parents of the proband.
- Family 5: segregation of 3 ultrarare variants (*WDR26* 1:224411450-C>T, *PAXBPI* 21:032769808-A>T, *CLDN5* 22:019523698-C>T, hg38) identified in proband of family 5 was performed confirming that the *PAXBPI* and *CLDN5* are maternally inherited while the

WDR26 is paternally inherited.

Taking together, both analysis support the paternity/maternity for both families underscoring that the identified variants are truly *de novo*. A sentence has been added to the manuscript referring to the new supplemental Table S4.

Lastly, we have added the pedigree of a new patient (new Family 6) who was found to harbor the same variant p.L668S of Family 5 that resulted maternally inherited.

Updated pedigrees of all families are available on Supplemental Figure 1.

- The authors wait until the discussion of this report with saying that the p.N840S variant may be mild or possibly benign. Can the authors please state that this is a variant of uncertain significance earlier in the report (e.g. in the last section of ‘Identification of WDR44 variants’ on page 8)? The authors should also add to this section that while the conservation is high for c.2519A>G, the REVEL score is in the tolerated range. The score for 2003T>C (p.L668S) is in the borderline damaging/uncertain range at 0.6299; it would be worthwhile to also mention that in the main text of the report. Please, feel free to also add that REVEL scores for the other variants are all in the damaging range. Please note that REVEL is written as RELEV in Table S2.

Response: We thank the reviewer for this suggestion and we have commented on the REVEL score for the variants as suggested at the end of the ‘Identification of WDR44 variants’ section, underscoring that the p.N840S is predicted to be tolerated. We have also corrected the spelling mistake in the supplemental table 2.

- The nonsense variant c.2197C>T is not located in the last or penultimate exon of WDR44. It is expected to lead to nonsense mediated decay and predicted to lead to an absent or an abnormal protein. The authors say the following about this variant ‘ The nonsense variant in exon 16 introduces a stop that is predicted to truncate the protein at the residue 733 in WD5’ on page 8. I think they need to explain on page 8 that the nonsense variant could lead to both NMD and/or an abnormal protein, which may or may not be truncation at codon 733. Rightfully, the authors do acknowledge the possibility of escape of NMD in their discussion. They say that this phenomenon is ‘not uncommon’ and refer to only a single paper where escape of NMD was reported as an example. Do the authors know what percentage of nonsense variants located in exons that are not the last or penultimate exon generally escape NMD? What is ‘not uncommon’?. I think of escape of NMD as something that is rare, but I might be wrong... If possible, it would be great if the authors could explore RNA/protein expression in fibroblasts of the patient with the c.2197C>T nonsense variant to solidify their hypothesis that the c.2197C>T variant is associated with an expressed abnormal WDR44 protein that leads to a gain-of-function effect.

Response: We thank the reviewer for this suggestion. During the revision period our clinical collaborators were able to culture fibroblasts from patient p.R733* (c.2197C>T). However, due to the ongoing Russia-Ukraine conflict it has not been possible to ship cells out of Russia for

testing. Our collaborators were able to perform QPCR expression analysis on the patient cells and showed that the nonsense variant mRNA levels were significantly reduced. In the revised manuscript we now include mRNA analysis of patient and control cells which indicates that mRNA levels are partially reduced in the patient (data included in Figure 2b of the revised manuscript). These results support a conclusion that the R733* nonsense variant is sensitive to nonsense mediated decay (NMD). Our collaborators have tried to examine protein expression but have observed conflicting results which we believe is related to issues with different WDR44 antibodies they are able to acquire. Ciliation assays are beyond the scope of this group at this time. We hope in the future to be able to perform additional studies on these patient cells. Based on these results we have removed comments related to this protein escaping NMD.

- Figure 7c. If the authors have access to patient-derived cells (other than N840S) and WDR44/RAB11 antibodies, what do endogenous localizations look like in patient-fibroblasts versus controls? Are WDR44 and RAB11 expressed in human fibroblasts?

Response: We thank the reviewer for this suggestion. As suggested by the reviewer we examined RAB11 protein expression in patient fibroblasts and did not observe any differences in variant and control patient cells. We have included this data as new Figure S7. By way of clarification to the reviewer question, WDR44 protein expression in human fibroblasts was shown in patient and control cells in the previous submitted manuscript (Figure 2a).

We did attempt to examine effects on WDR44 localization in patient cells. As we have observed in our previous work in RPE-1 cells (Walia et al., 2019) we could not detect a specific endogenous WDR44 signal after testing all available commercial WDR44 in patient fibroblasts using PFA and alcohol fixation. We observe a similar IF signal with WDR44 Ab in control and patient fibroblasts.

We also examined RAB11 localization using antibodies kindly provided by Dr. Jim Goldenring, and while we could detect characteristic RAB11 localization we did not observe an obvious disruption in RAB11 localization in control vs patient cells. This may be due to a high degree of cell to cell variability observed with RAB11 localization (more diffuse small punctate cellular staining and larger punctate structures). It may not be surprising that RAB11 localization appears normal given the model we proposed in our previous paper that the ciliogenesis pathway occurs via a switch from WDR44 effector binding to FIP3-Rabin8 binding associated with a recycling endosome pathway associate with the centrosome. RAB11 is associated with other cellular trafficking pathways involving several different effectors which could be independent of this pathway and would therefore complicate assessment of changes in RAB11 localization due to WDR44. As we are not able to conclude if endogenous RAB11 localization was affected in patient cells we have not included this data in this manuscript.

It would be very compelling if Figure 7c findings are verified with endogenous data.

Response: We thank the reviewer for the suggestion to perform IP studies in patient fibroblasts. We attempted both WDR44 and RAB11 immunoprecipitation studies using control fibroblast cells but we could not observe interactions even with WT WDR44 human fibroblasts and therefore we

did not include this negative data. This is consistent with our previous published work in RPE-1 cells where we also could not perform endogenous IPs (Walia et al., 2019). We also attempted IPs with variant patient cells but could not detect interactions with RAB11. Overall, this and our previous published results could be explained by a pathway limited interaction associated with ciliogenesis initiation. Notably, mother centriole uncapping ciliogenesis initiation is associated with a 300nm ciliary vesicle structure which would only require a small number of vesicles to assemble this structure. In our experience it is also not uncommon to fail to detect endogenous RAB11-effector binding due to potential GTPase activity on RAB11 which could disrupt RAB11 effector interactions detection during the IP.

The authors should briefly specify the most common clinical features in the main abstract of the paper and at least mention ID, hypotonia, characteristic facial features as all patients had at least these features along with variable other characteristics.

Response: We thank the reviewer this suggestion. The abstract has been modified as follow :
“Patient phenotypic spectrum includes developmental delay/intellectual disability, hypotonia, distinct craniofacial features and variable presence of brain, renal, cardiac and musculoskeletal abnormalities.“

Please do not overstate your conclusions. I like that data is summarized per section, but the detected variants do not all have similar effects and therefore it is important to phrase your conclusions more carefully. Some examples: Page 13: Thus, these results indicate protein abundance of WDR44 variants is also affected in zebrafish >> Thus, these results indicate protein abundance of various WDR44 variants is also affected in zebrafish. Page 13: Together, these results recapitulate the pathogenicity of the human WDR44 variants associated with ciliopathy phenotypes >> Together, the majority of WDR44 variants recapitulate the WDR44-associated human features in zebrafish. Page 14: ...suggesting that WDR44 variants exert GOF in repressing ciliogenesis >> ...suggesting that various WDR44 variants exert GOF in repressing ciliogenesis.

Response: We thank the reviewer for this suggestion and have made the corresponding changes suggested by the reviewer related to comparison of human and fish variants observations.

Minor:

- The use of the word ‘mutations’ (e.g. page 2, 6, probably others) is out of date and should be replaced with ‘variants’ or ‘pathogenic variants’ (whatever is relevant at each section) throughout the manuscript.

Response: We thank the reviewer for pointing this out and we have replaced references using ‘mutations’ throughout the manuscript.

Page 2: .. encoded by X-linked gene > ...encoded by the X-chromosomal gene WDR44 (OMIM *301070).

Response: We thank the reviewer for this suggestion and have replaced in the text as suggested.

Page 6: ... Ciliopathy-associated missense variants show enhanced proteasomal degradation... >> WDR44 containing ciliopathy-associated missense variants is subject to enhanced proteasomal degradation

Response: We thank the reviewer for this suggestion and have replaced in the text as suggested.

The resolution of all pedigrees in Figure S1 is poor and should be replaced with images with improved resolution. The text is difficult to read.

Response: We apologize for the resolution issue related to compression of the pdf. We have provided higher resolution figures of the pedigrees in the revised manuscript.

Figure 1 and S1, the images of the patient with the p.D648G variant seem squished?!

Response: We thank the reviewer for this observation. We have replaced this image.

In protein nomenclature there should not be any spaces between the p. and the remainder of a variant, e.g. p. S764F should be p.S764F. Please check nomenclature throughout the manuscript for all variants in the main manuscript, legends and supplemental documentation.

Response: We thank the reviewer for catching this mistake. We have fixed the naming nomenclature to be consistent between sections.

One could add in the manuscript that the Z-score of the WDR44 gene in gnomAD is 2.95, which provides some support for intolerance of missense variation in this gene. Null variants in this gene are certainly selected against.

Response: We thank the reviewer for this suggestion and we have added the Z score in the "Identification of WDR44 variants" section.

Page 7: The authors say that they assessed WES trio in their first patient (p.S764F) for LP/P variants in known disease genes, and that they subsequently focused on ciliary genes. Did the authors also use alternate prioritizations? For example, all genes known to be expressed in brain development? Were there any other candidate genes found?

Response: For the index patient of family 1 we used a standard approach looking at non-synonymous-exonic and splicing variants with a minor allele frequency (MAF) ≤ 0.001 in the gnomAD database. We took into consideration all possible inheritance (heterozygous, homozygous, compound heterozygous and X-linked). We first prioritized variants in the Online Mendelian Inheritance in Man (OMIM) morbid genes. No candidate variants were flagged in any OMIM genes.

Of note WDR44 was not listed as an OMIM gene (creation date 02/15/2022) at the time of the exome analysis in January 2020. When we extended the analysis to all genes, the WDR44 variant appeared to be the most probably candidate for our patient considering several prediction tools for the specific variant (Supplemental Table 2) and the gnomAD constraint metrics (Z score 2.95, intolerant to missense variants). We have added the following to the results section. ‘WDR44 was recently listed as an OMIM gene (creation date 02/15/2022) and is highly constrained for missense variation (gnomAD Z-score: 2.95).’

We did not prioritize variants for expression in brain development considering that expression can significantly vary from embryonic to postnatal development for a specific gene. However, manual inspection looking at expression in GTEX and Protein Atlas databases and literature review was done for all ultrarare possible candidate variants. To clarify this better in the results section we have added this sentence ‘A comprehensive analysis looking at all genes and considering all possible mechanisms of inheritance did not identify other candidate variants. In this regard, we have added the rare variants identified in patient of family 1 in the supplemental material (Supplemental table 3).

In the case of other patients our collaborators for family 2-9 confirmed that no pathogenic/likely pathogenic variants were identified in known disease-causing genes nor candidate variants in other genes were flagged besides *WDR44*.

It is somewhat surprising to me that unilateral multicystic kidney findings were noted in patient with p.S764F because as far as I am aware renal cysts tend to be bilateral in ciliopathies. Are there other reports of cilia dysfunction and unilateral renal cyst findings? How do the authors explain this? Do they suspect the other kidney may also develop cysts in time? Was there anything remarkable in the imaging of the other kidney?

Response: We agree with reviewer that ciliopathies are classically associated with bilateral multicystic kidney. However, unilateral renal cysts have been already reported like for instance in Joubert syndrome and Bardet Biedl Syndrome. We have added a sentence and the reference Grochowsky and Gunay-Aygun 2019 (PMID: 31763176) to the discussion. We are unaware if in these instances the unaffected kidney later becomes cystic. However, we have noted in discussion it is possible that patients could develop cysts as they get older.

Although unrelated to cystic development, we provide an update in the revised manuscript that the patient from family 5 with previously reported kidney hypoplasia has developed stage 2 chronic kidney disease (CKD2). A recent follow-up showed that also patient of family 3 is developing renal failure (Supplemental table 1). We have updated the manuscript with this information.

S4f. Is there a Wdr44 antibody available? If so, can the authors show where/when endogenous Wdr44 expresses in zebrafish?

Response: We thank the reviewer for this suggestion and we also hoped it would be possible to detect the fish WDR44 protein. Unfortunately, the antibodies available did not detect the endogenous zebrafish protein which is why HCR was used.

Page 14: ... In contrast, morphants expressing WDR44 variants, except for D648G and L668S, caused defective ciliation in the neuromasts, olfactory placodes and pronephric ducts. >>> In contrast, morphants expressing WDR44 variants, except for D648G, L668S and N840S, caused defective ciliation in the neuromasts, olfactory placodes and pronephric ducts.

Response: We thank the reviewer for catching this omission and we have changed this in the manuscript.

Figure 7A clearly shows increased RAB11A binding in the majority of mutants compared to wild type. The authors state that the same can be seen in Figure 7B with endogenous RAB11A, but RAB11A binding seems somewhat reduced for G782C (maybe because of overall low GFP-WDR44 expression in WCL?). Can you please compile a similar graph as shown in Figure 7A: it may help to interpret Figure 7B?

Response: We appreciate the reviewer's comment. Because variant studies were performed at different times we have repeated this experiment with all the variants analyzed at the same time for 3 independent experiments so we could quantify the results as requested by the reviewer. Because we have not exogenously expressed RAB11 and VAPA and to avoid any effects of the endogenous wild-type WDR44 protein on our IP results we performed this experiment in RPE-1 WDR44 KO cells instead of 293T cells. As suggested we have performed densitometry for both RAB11 and VAPA bindings. The results for RAB11 binding in Figure 7B are consistent with 7A, and as noted by the reviewer G782C displayed statistically lower binding than the other variants, except N840S. As the reviewer noted this variant appears to be expressed at lower levels than the majority of the other variants.

Similarly, binding with VAPA seems to be lowish in various mutants? Is that just artificial variation of the assay or is there is an additional variant effect?

Response: We have performed densitometry on VAPA binding and confirm that all the missense variants except 648 have no statistically significant difference in VAPA bindings. Based on these results we have added a statement to the results and discussion indicating that the 648 variant and the 733* have increased interaction with VAPA which could be associated with WDR44 variant gain of function in diseases.

The authors state in the discussion that cerebral and cerebellar white matter changes may

be due to ciliary defects. This is a possibility, but it also possible that this phenotype is due to disruption of an extra-ciliary function of WDR44. The authors say that WDR44's function is not limited to cilia earlier in the manuscript and in the final conclusion as well, but I think it is worthwhile to also mention this in the white matter discussion because that phenotype does seem rather unique.

Response: We agree with the reviewer to consider other mechanism besides ciliary dysfunction that may account for non-classical ciliary-related features. The following sentence has been added in the manuscript discussion: "However, it is also possible that this neuroimaging pattern might be due disruption of an extra-ciliary function of WDR44 that is yet unknown."

Responses to Reviewer #2:

The manuscript by Acogli et al describes a new gene-disease association with mutations in WDR44 in 10 patients presenting with complex neurodevelopmental and multisystemic disease. The authors demonstrate that the variants, which are mostly missense variants, lead to decreased expression levels of the mutant WDR44 protein through proteasomal degradation, but suggest that the remaining WDR44 variant protein acts through a gain-of-function mechanism with increased WDR44-RAB11 binding, which in turn affects ciliogenesis by decreasing the ability of RAB11 to release the CP110 capping required in the first steps of ciliogenesis. Overall, this represents an interesting story; the manuscript is overall well written and structured. However, some parts of the results are not as compelling as they could be: while the association between the described WDR44 variants and human disease appears to be strong, including the effect of the variants on binding to RAB11, the actual direct link to cilia appears less convincing. The authors claim that this is a ciliopathy based on the patients' clinical phenotype, on the zebrafish data and on the ciliation experiments in cell lines. I would argue however, that (1) the phenotypic constellation of the patients described is not really typical for bona fide ciliopathies (see details below); (2) the zebrafish data is moderately convincing due in part to some technical problems (see details below); (3) the cell culture experiments are mostly performed with overexpression experiments in lines that still contain endogenous protein (which is not addressed at all) and fibroblasts were available only from 3 patients; moreover, the data on ciliation are somewhat confusing/show different results between the patients and between fibroblasts and overexpression experiments in various cells types, which weaken the conclusion that the WDR44 variants cause the phenotype through a direct effect on ciliogenesis and that the patients' disorder is a "first order" ciliopathy.

Response: We thank you the reviewer for their comments and we have addressed all three main points below. In our revised manuscript additional patient, human cell and zebrafish data is provided, and clarification is provided in the response to the reviewers' points, that support our conclusion that patients with WDR44 variants in the WDR domain have a ciliopathy-related disorder due to gain-of-function in binding to RAB11 important for regulating ciliogenesis initiation

Detailed main points:

(1) patient phenotypes as an indication for a ciliopathy:

the patients do present with pleiotropic disease, which does affect some of the organ systems that are also commonly affected in ciliopathies, but the way these organs are affected is not really typical or specific for the classical ciliopathies (for example, while there is musculo-skeletal involvement in these patients, this is not a patterning defect with polydactyly as typical for ciliopathies or a skeletal dysplasia as in the short rib polydactylies or other skeletal ciliopathies; brachydactyly, syndactyly or scoliosis are all found in many other non-cilia-related disorders, probably even more often than in ciliopathies; only one patient has cysts in one kidney and one patient has a nephritis, so renal involvement is also not prominent in this cohort (other renal symptoms such as nephrolithiasis or renal hypoplasia are not cilia-related phenotypes); the CNS anomalies mostly appear to affect white matter, which is not typical for ciliopathies, and in a few cases the corpus callosum, but this is also seen in many non cilia-related syndromes). So all in all, calling the patients' phenotypes "ciliopathy-related" per se and using this as an argument to support the link between these variants and a direct link to ciliary defects seems a bit of an over-interpretation (they might be compatible with ciliary dysfunction but cannot be seen as typical for this). Other syndromic disorders have been shown to have an indirect effect on cilia and ciliogenesis and can be described as "second-order ciliopathies". This possibility should at least be mentioned. Depending on the extent to which the authors can address the following comments, I think that the authors may have to be more conservative in their overall interpretations.

Response: We thank Reviewer for pointing out this observation and suggestion. As suggested by the reviewer we have added the following to the discussion 'Although WDR44 variants have pleiotropic effect with multi-organ dysfunction in several patients, classical ciliopathy-related features may be more subtle. Thus, the observed WDR44-related disorder might belong to the continuously expanding list of second-order ciliopathies, i.e. disease that are caused by pathogenic variants in genes encoding protein that are not localized within cilia but that have a role in cilium formation and function¹⁴

We have also modified the discussion as follows. A sentence has been added when we comment the white matter anomalies in the discussion: "However, it is also possible that this neuroimaging pattern might be due disruption of an extra-ciliary function of WDR44 that is yet unknown."

We have also removed the "ciliopathy-related" in several points throughout the manuscript to focus more on the severity of pleiotropic disease phenotypes for the patients disease. During the revision we were able to gather more clinical data for our patients which strengthen the association of this disease with ciliopathy including brachydactyly data, and II-III toe syndactyly (two recurrent ciliopathy features) in 6 and 5 patients, respectively (Figure 1,b,d, S1e, Supplemental table 1 of the revised manuscript).

Regarding kidney findings we are aware that unilateral renal cysts are less common in ciliopathies despite they have been reported (see added comment and reference Grochowsky and Gunay-Aygun 2019 (PMID: 31763176) in discussion). We have moved the finding of nephrolithiasis from

the main text to the supplemental table since it may be a possible unrelated feature. As mentioned in the rebuttal for Reviewer #1, It is noteworthy that 2 other patients developed chronic kidney disease: patient of family 3 who has nonimmune nephritis had increased creatinine and reduced eGFR at last evaluation consistent with initial stage of renal failure (CKD2) (Supplemental table 1). The manuscript has been updated accordingly. Patient of family 5 has as well CKD2 with kidney hypoplasia. Renal biopsy has not been performed in these 2 patients at this time which is needed to define the nature of renal failure.

We believe the revised results from zebrafish (expression analysis in new Figure 4a and cystic glomeruli in new Figure 4i and human cells (Figure 3 and 6) provides additional compelling support our conclusion that our WDR44 patients have a ciliopathy related disorder.

(2)zebrafish experiments:

One problem with the ZF experiments is that morpholinos were used to knock-down endogenous *wdr44*; it remains unclear why this approach was chosen (the rationale underlying the statement “To closer mimic the hemizygous state of WDR44 patients we used morpholinos (MOs) to reduce expression of the endogenous *Wdr44* protein” is hard to understand. For an X-linked gene the amount of protein is the same in a hemizygous male as for an autosomal gene in a homozygous individual as far as I know. So I am not sure what the advantage would be to titrate the amount of gene knockdown in this situation (plus, an actual titration is not described in this work). Moreover, the community now usually agrees that morpholinos should only be used with proper controls which include comparison with a mutant (see recommendations paper doi: 10.1371/journal.pgen.1007000). Particularly in this situation, where the authors then overexpress the variants, it would have been much cleaner to do so in a knock-out background. I can see the point that the morphants have no phenotype (see point below) so there is not much to compare to a mutant, but maybe this would be a good reason to see what happens when the gene is knocked-out

Response: We apologize for the lack of clarity in our rationale for the use of a morpholino knockdown approach to mimic the male patients hemizygous state. We have removed this statement and replaced with the following ‘Embryos were treated with *wdr44* morpholinos (MOs) to reduce expression of the endogenous *Wdr44* protein and were validated by examining a GFP reporter RNA containing the 5’ UTR target site of the MOs (Figure S5b).’ Our goal was to test WDR44 human wild-type and variants in a fish background without potential complications of endogenous fish *Wdr44* given that we know that the variant protein is expressed at lower levels than the wild-type protein. Importantly, we now provide new expression data confirming this effect on expression (Figure 4a). We appreciate the reviewer sees that there is no reason to compare the morphants to mutants given the absence of effects of the morpholinos. As suggested by the reviewer, we present new data showing that sequencing confirmed CRISPR/Cas9 knockouts of *wdr44* lacks ciliation effects on the organs examined in F0 embryos (Figure S5h). A complete knockout would have been much cleaner in hind-sight but given the current time and resource constraints this was not possible. We have also added additional ciliation results for the second MO showing no observable effect on ciliation (Figure S5g) which is consistent with MO1.

The authors also provide insufficient information on the morpholinos used: are these ATG morpholinos or splice morpholinos? In the latter case, a splicing study would have been nicer

to show the effect of the morpholino (rather than the effect on GFP expression of a co-injected construct).

Response: We apologize for this omission in the methods section, both are non-overlapping ATG morpholinos; MO1 was previously described in Walia et al., 2019 and MO2 was custom designed by Genetools. We agree with the reviewer that a splice morpholino would have been a good alternative approach. In the present revisions, we chose to further confirm our results using a CRISPR approach as was suggested by the reviewer for the reasons described above.

It is also surprising that the morphants display no phenotype at all (how closely were these analyzed? Is there no effect on cilia at all and how does this fit with the knock-out data in cell lines which exhibit higher ciliation rates?)

Response: We agree with the reviewer in that we did not know how an increased ciliation in human cells would translate into the zebrafish. We have now used three independent genetic (2 MO and CRISPR/Cas9) tools to knock-down endogenous *wdr44* and found no ciliary defects in the organs of interest. One possibility is that premature ciliation occurs in some cells, which to our knowledge has not been described in fish embryos and was not the focus of this work.

Moreover, the WB shows that none of the mutant forms of the protein is detected in zebrafish lysates except for N840S (figure S4e), which questions how the effect can be caused by a mutant protein which is not detectable (the possibility that the variant affects the antibody epitope is not supported by the fact that the same mutations are detected in human fibroblasts). In figure 4i it is also rather hard to understand how this was quantified and how the GFP results fit with the WB data shown in figure S4e?

Response: We appreciate the reviewer's concern about phenotypes being caused by proteins we could not detect by traditional western blotting. We do believe the GFP-tagged proteins demonstrated that these variants could be expressed, but of course these were not the versions of WDR44 wt and variants assessed for phenotypes and ciliation. We are happy to report that we can detect expression of all patient variants using dot blot (Figure 4a) and the variants all show lower expression than the WT protein in fish embryos. This result is consistent with our previously described expression data (we removed in favor of new Figure 4a). We appreciate the reviewer's question about epitope detection. We have now clarified that we used the same antibody to detect the variants as was used in human cells (Bethyl). This antibody was raised against an epitope between amino acid 100-150, which is not affected by patient variants and shows a similar protein detection as with GFP antibodies.

Finally, and most importantly, the phenotypes shown after overexpression of the mutant Wdr44 forms can hardly be considered "typical ciliopathy phenotypes":

Response: We thank the reviewer for this suggestion. We rephrased this sentence in the manuscript and replaced the word "typical" by "cilia-related".

The kidney cysts are not visible on several of the images in figure 4h (why was this looked at 6dpf rather than 3dpf where the MO and the mRNA are likely still active while at 6dpf their effect could be gone?), and the remaining phenotypes look in fact just as similar to non-specific toxicity effects of mRNA/MO injection (heart edema, small heads, crooked fish).

Response: These data were removed and replaced by an analysis of the pronephric tubules at 3 dpf using the Tg(*wtlb*:EGFP) model (shown in Figure 4i). This analysis better shows evidence for the presence of glomerular cysts formation at this earlier developmental timepoint. These effects show the specificity of ‘cilia related’ effects observed in this assay and on ciliation for WDR44 variants.

A recent paper with comprehensive comparison of multiple ZF ciliopathy mutants has not found any of these features (doi: 10.1242/dmm.049568). I acknowledge the fact that the WT mRNA did not appear to cause any phenotypes, which would argue that the mutant forms have an additional effect, but unfortunately, I’m not sure this suffices to claim specificity of a gain-of-function effect of the mutant protein (which is not detectable by WB) on cilia-related phenotypes. The cilia-defects shown in figure 5 are also not so convincing: in several images, the actual organ analyzed (neuromast, olfactory placode) appears very underdeveloped, likely in line with the fish being rather sick, and so it’s difficult to interpret an effect on cilia in a malformed organ.

Response: As noted above we now provide dot blot analysis confirming expression of patient variants in fish which addressed concerns about gain-of-function effect. Mutant embryos phenotypes persisted beyond 3 dpf (Figure 4h), suggesting that they were not caused by developmental delay.

It is also unclear how the quantifications were performed (yes/no cilia anomalies per embryo?)

Response: We thank the reviewer for pointing out this omission. For clarification, we added a citation of our previous publication (Lu and Insinna et al., NCB 2015), which described our method of analysis in the method section. Organs with absent or strongly reduced cilia numbers were quantified and this data was used to calculate the percentage of normal versus total number of organs observed. This description was added in the method section.

Ideally, the authors should express less *wdr44* mRNA in a stable mutant knock-out background to make these results more compelling. This does represent a substantial additional amount of work, but could substantially strengthen the link to cilia. I would recommend at the very least to characterize the morphant phenotype better and to compare it to F0 crispants to have an alternative method for interfering with the endogenous gene.

Response: As mentioned above we included new ciliation data for F0 crispants as suggested by the reviewer. Together, these results support our previous analysis that depletion of WDR44 does not cause effects on the target organs we have investigated.

Most importantly, I would recommend decreasing the dose of injected mRNA so as to have embryos that look less sick and analyze the cilia in these in a more stringent manner (actual quantification of ciliation in selected organs).

Response: We agree that reducing mRNA levels (protein expression) is expected to reduce the phenotypes observed on embryos. However, as we now show that all WDR44 variants are expressed at lower levels than the wild-type protein we are concerned that it will not be possible to detect these variant proteins, which the reviewer had already mentioned as a concern. Embryo survival until 6dpf (Figure 4h) supports tolerance of mRNA treatments for embryos. Stringent mRNA titrations were performed to establish injection levels comparing to wild-type.

(3)Cell experiments:

The most compelling data concerning the link between novel gene variants and ciliation often comes from patient fibroblast data, since this represents the closest situation to the disease state without additional interference; unfortunately these are generally hard to get and indeed, in this study, lines were available only from 3 patients. What is a little disturbing, is that the ciliation results differ between these three patients (while the significance of N840S is unclear as also discussed by the authors, and this line may be considered separately from the others throughout the paper, the other two lines should behave more similarly). Some degree of difference in severity is of course acceptable, but it is rather disturbing that only S764F can be rescued with respect to ciliation and not D648G. This substantially weakens the claim that WDR44 variants cause a phenotype through a defect in ciliation. Then the authors drop the D648G line in the following experiments. So in the end, the only really strong and consistent data supporting an effect of WDR44 variants on ciliation is on one variant (S764F).

Response: We agree with the reviewer about the potential for using patient cells. The fact that we do not see rescue in the D648G cells does not rule out this variant affecting ciliation in these cells. To clarify these results we have reworded to ‘To determine if reduced ciliation can be rescued by wild-type WDR44, we examined ciliation after stably expressing GFP and GFP-WDR44 in fibroblasts (Figure 3b)’ As discussed below we have added full analysis of the D648G cells (Hh and CP110, Figure 3e and 6a respectively which show a consistent effect from reduced ciliation on CP110 uncapping and Hh signaling. One notable observation we made in new data in Figure 7B is that the D648G variant binds more strongly to VAPA suggesting this variant and S764F variants do not behave the same way which could be associated with disease presentation and the absence of rescue observed.

The authors use a variety of assays in a variety of cell lines which give somewhat different results that make things rather confusing: 2 patient fibroblast lines have no cilia without

serum, but only one line has normal ciliation with serum (a); the third line (N840S) has no effect on ciliation but has shorter cilia; expression of myc-tagged WDR44 variants in 293T cells (where endogenous protein is not analyzed/interfered with) shows no effect on ciliation rate but results in shorter cilia for all variants. The authors then go on and make RPE1 KO lines for WDR44 and analyze CP110 capping after expression of selected WDR44 variants (why only some?) in the RPE1 KO lines (and in only one patient fibroblast line).

Response: We apologize for any confusion the use of several cell systems which in some case was done out of necessity for what was being studied. We also feel the use of different cell types helps demonstrate that these variants have similar effects on ciliation in different cell types.

The reason 293T cells were used for ciliation analysis is that they can more robustly undergo ciliogenesis in transiently transfection experiments when examining transfected cells than is observed in RPE-1 KO cells. In contrast, CP110 uncapping can be monitored appropriately following transient transfection of GFP in the RPE-1 WDR44 KO cells.

We thank the reviewer for the suggestion to include patient fibroblast cell data (Hh and Cp110 uncapping; Figure 3e, 3f, 6a) and we have examined CP110 uncapping with all variants (Figure 6c). These results are consistent with reduced ciliogenesis initiation by WDR44 variants.

Then the authors go back to analyzing RAB11 localization in 293T cells. Why they go back to 293T cells is unclear. In fact, now that the authors have made a KO line lacking endogenous WDR44, the results could be much more compelling if all the variants were expressed and analyzed in this null background: ciliation for all variants could be assessed in this setting, in parallel with the CP110 capping, the HH assay and the RAB11 assay.

Response: There may have been some confusion about the question about ‘going back to 293T cells’ to examine Rab11 localization. These studies were performed in RPE-1 WDR44 KO cells (Figure 7c). In terms of the Hh assay we agree this would have been an ideal to examine Hh signaling in WDR44 KO + rescue cells, however, in our hands RPE-1 cells do not show Hh signaling. New Figure 7b (analysis of endogenous RAB11 and VAPA) binding has been repeated in RPE-1 WDR44 KO cells and the previous 293T data has been removed.

Consistent results with such a more unified set of experiments would substantially strengthen the claim that these WDR44 variants cause disease through an effect on ciliogenesis.

Response: This point was addressed in response to the reviewer’s comments and we believe we have included new data and experiments and clarified description of results that strengthen the conclusion that WDR44 variants have affected ciliogenesis.

Similarly, it should be possible to look at CP110 and RAB11 in the patient fibroblast lines that are available (all 3), in addition to the ciliation analysis.

Response: We thank the reviewer for this suggestion. We have added CP110 data for all three patient cells as mentioned above (Figure 6a). Reviewer #1 also asked about Rab11 localization in patient fibroblasts. We provide a blot showing that RAB11 expression is not affected in patient cells (new Figure S7). However, as noted in our response to reviewer #1 it was not possible to determine if RAB11 localization was affected. We hypothesize that WDR44 variant levels are not high enough to cause observable changes to Rab11 localization as we can see when we overexpress variants in RPE-1 WDR44 KO cells.

Comments per figure:

Figure 1:

Some of the patient figures appear to be distorted (panel 2 and probably 1 of figure 1a and panel b of figure S1). The image resolution was too low to be able to increase the size of the MRI images sufficiently to actually see things well enough.

Response: We thank the reviewer for this comment and have corrected as described in response to reviewer #1.

Figure 2:

2c: why are the stats not indicated for the comparison between WT and the truncating variant? If this is not significant, it might be better not to mention in the text that there is a trend for an increased expression?

Response: Yes statistics was performed and were not significant. We have reworded this in the results to ‘while the nonsense GFP-tagged R733* variant was expressed at similar level as the wild-type tagged protein.’

2e: The control with MG (panel e) also appears to show a much stronger band, which is not reflected in the quantification graph?

Response: We thank the reviewer for pointing this out. We reexamined the GFP-WDR44 control expression levels using a lower exposure blot and replotted the data in Figure 2e. We do not see a significant increase in the control between untreated and MG132, although not unexpectedly the control shows a trend for more protein, but the fold change is less than what is observed with the affected variants.

2c compared to 2e: in 2c, several variants show very decreased expression but in 2c with chloroquine (almost) no difference?

Response: We apologize if there is any confusion Figure 2c does not have a chloroquine treatment. In figure 2e we observe all GFP-WDR proteins to show increased expression with MG132 treatment compared to control (untreated) and chloroquine treatment. GFP-WDR44 WT and variants have similar expression levels in Figure 2c and 2e.

Figure 3:

3b: why are ciliation levels not restored in D648G but only in S764F? If dominant negative / gain-of-function effect, than overexpressing the WT form should not rescue either, but if loss-of-function effect, than WT should rescue both?

3d-e: no data on D648G?

Response: We addressed this comment above in Reviewer #2 major section

Figure 4:

Quantifications are yes/no per embryo to reach the % shown?

Response: We addressed this question above and have clarified in the manuscript.

The term “scoliosis” is not really appropriate for a zebrafish larva that has no skeleton at this point. Body curvature would be a more descriptive and conservative term.

Response: We thank the reviewer for this suggestion and have changed ‘scoliosis’ to ‘body curvature.’

4h: cysts are not visible in several panels (when enlarging, the resolution is insufficient)

Response: As noted above we now include new data using Tg(*wt1b*:EGFP) zebrafish model to show affected glomeruli and have removed the bright field images.

4i: unclear how the +, ++ and +++ were determined; not obvious from the images

Response: We have removed the GFP-WDR44 variant expression data in favor of new dot blot expression analysis (Figure 4a)

Figure 5:

Unclear how the quantifications were decided (as above)

Response: We addressed this comment above

5a shows no phalloidin

Response: We thank the reviewer for catching this mistake. We have removed phalloidin from Figure 5a and the accompanying legend.

Figure 6:

6a: why is there no data shown on the fibroblast line of patient D648G?

Response: As suggested we have added this data to new Figure 6a.

Figure 7:

7b: I am not sure that the difference between RAB11 and VAP is very compelling; the text in the corresponding paragraph is also written in a somewhat confusing manner.

Response: We have repeated these experiments and provide new data in Figure 7b in order to quantify the observed differences statistically.

Discussion:

In the discussion the emphasis on the “ciliopathy phenotypes” appears too strong (see comments above). Some parts of the discussion also are bitt too much of a repetition of results with detailed re-discussion of each variant and reference to figures again; maybe this should be summarized a little more.

Response: As suggested by the reviewer we have modified the text throughout the manuscript to address emphasis on ciliopathy phenotypes. As noted in above comments to the reviewer’s suggestions we have added new supporting data and clarified results that implicate disease connections to cilia function. We also thank the reviewer for the suggestion and we have modified the discussion where appropriate to reduce repetition of results.

REVIEWERS' COMMENTS

Reviewer #1 (Remarks to the Author):

Thank you for the additional experimentation. The evidence for a gain of function mechanism is still incomplete in my opinion. Prior to acceptance, the claim for gain of function mechanism needs to be significantly weakened throughout the report, particularly in the title (gain-of-function wording should be removed), abstract and discussion. I am not opposed to presenting the gain of function findings as a model for disease mechanism (and I certainly find your findings interesting!), but in my opinion the authors just cannot rule out the possibility of loss of function mechanism and that should be clearly worded in the report. The gain of function hypothesis is a sexy idea, but it is not a common mechanism in ciliopathies and after reading the revision I still feel that I am not convinced. If the authors want to publish their report without additional experimentation they need to clearly acknowledge the possibility that the mechanism of disease could also be loss of function for the following reasons: 1) There is a strong reduction in reduced transcript level of p.R733* in cells derived from this patient (as well as various other patients) and the authors acknowledge there is sensitivity to NMD for this patient in the revised manuscript. There is no evidence shown of protein expression in patient fibroblasts, which could have provided compelling evidence for gain of function effects. The authors only refer to exogenous expression of p.R733*, but I consider that poor evidence; 2) The pLI score of WDR44 is 1 (gnomAD) and suggests this gene does not tolerate loss of function; 3) The 1.9Mb deletion reported by Pavay (PMID: 27194972) involves multiple genes and exon 1 of WDR44 in a patient with an overlapping phenotype. I believe this is a loss of function variant as the first downstream ATG of WDR44 is in exon 3 and this ATG is not in frame. The authors rightfully state that other genes were also deleted in that 1.9Mb deletion and that the evidence for pathogenicity for WDR44 is therefore limited. While that is true, haploinsufficiency of WDR44 is of course suspicious, particularly after reading about the detection of a nonsense variant in your report; 4) The expression of endogenous mutant WDR44 is lower compared to controls in patient-derived fibroblasts (of various individuals with convincing variants anyway) and mutant GFP-tagged WDR44 proteins is also lower compared to wild type WDR44 expression, which is generally consistent with loss of function; 5) The authors did not assess knockout models (fish/fly/mouse) for learning impairment; 6) The phenotype of the p.R733* patient is more severe compared to missense, which would make sense in context of loss of function.

The authors should mention in their report that they cannot rule out a loss of function mechanism of disease and that future experimentation (such as assessing neurological function and learning/memory in knockout models, functional assessment of cells from other patients with nonsense variants) is required to confirm their proposed gain of function model. Also, please change the title to "Variants in the WDR44 WD40-repeat domain cause syndromic intellectual disability possibly caused by impaired ciliogenesis initiation." The last sentence of the abstract should be "This study provides new insights into WDR44 WDR structure and characterizes a new syndrome that could result from impaired ciliogenesis. Although a gain of function model has been proposed as a mechanism of disease based on this work, we cannot rule out that loss of function plays a role in the etiology of this disorder at this time." In the first paragraph of the discussion, please change to "Based on comprehensive in vitro and in vivo experiments we propose a molecular mechanism of disease wherein WDR44 variants within the WDR result in a gain of function in negative regulation of ciliogenesis through increased RAB11 binding activity. Interestingly, we find that more severe disease causing missense have a destabilizing effect on the variant protein likely resulting from structural defects in the WDR. Moreover, WDR patients variants disrupt interdomain interactions that are associated with regulating binding to RAB11. While this gain of function mechanism is an attractive model, we cannot rule out that loss of function of WDR44 contributes to the ciliopathy-like phenotype in our patients, because [include reasons]". In the p.R733* discussion, please extend the discussion to "Unfortunately, further studies with R733* protein from patient fibroblasts were not possible at this time and therefore we cannot rule out loss of function as a mechanism of disease". In the last paragraph of the discussion, please also address that loss of function (using those words) cannot be ruled out as a mechanism of disease.

p.R733*

Endogenous protein evaluation in patient-derived fibroblasts would have been very helpful, but I understand that it was not possible despite best efforts. The exogenous expression evidence is weak; please acknowledge that. Also please adjust this sentence on page 8: "The nonsense variant in exon 16 introduces a stop codon that is predicted to truncate the protein at the residue 733 in WD5 (Figure 1g)". With the knowledge you now have regarding sensitivity to NMD, I suggest to reword that to "The nonsense variant in exon 16 introduces a premature stop at codon 733 of the encoded protein. This is expected to lead to an absent protein and/or a protein that is truncated at residue 733 in WD5 (Figures 1g, 2b)".

p.N840S

Thank you for adding a comment relating to the p.N840S variant on page 8 that says "...with the exception of p.N840S that was of uncertain significance". It is not completely clear from this sentence if the in silico tools render uncertain predictions or if the variant is classified as a variant of uncertain significance. I suggest to reword to "...with the exception of p.N840S, which we classified as a variant of uncertain significance.". It is important to use clear language as laboratory/clinical geneticists, who will be using your report in the future as a reference, will be looking for variant classifications and whether or not a variant is considered likely causal or not by the authors. I also recommend adding a column with classifications in Table S2.

Pedigrees

Thank you for conducting additional segregation analysis, which has strengthened the genetic portion of your report. Regarding Family 9: please, either remove cousin III:3 from the pedigree as the current figure suggests that 'the variant does not segregate' or keep III:3 in the pedigree and grey the square and add a reference in the legend explaining that detailed phenotypic information is lacking but it is thought that the phenotype includes/may include mild speech delay, febrile seizure and ADHD.

Nomenclature

The authors corrected some nomenclature but there are still inconsistencies, e.g. N840S in some sections versus p.N840S in others. Please check all variants (not just N840S) and be consistent where you can, thereby applying HGVS guidelines throughout the manuscript (<https://varnomen.hgvs.org/>).

Variant Interpretation

Thank you for adding the Table S3 with variant prioritization data for family 1. While I appreciate the addition, the table is not very easy to review in its current form. It would help to have information such as RefSeq/cDNA/protein nomenclature and zygosity in the start of the table (rather than somewhere in the middle). I also suggest to add a column with brief comments for each variant in the start of Table S3 why it was (de-)prioritized (and possibly ACMG classification). The authors claim they reviewed GTEX expression, protein atlas database and literature but that information is not provided.

Also regarding variant prioritization, the authors state in their revised report "A comprehensive analysis looking at all genes and considering all possible mechanisms of inheritance did not identify other candidate variants." I think that may be formulated too strong. The authors should consider changing that to "A comprehensive analysis looking at all genes and considering all possible mechanisms of inheritance did not identify stronger candidate variants.". The following sentence is also too strong "WDR44 was recently listed as an OMIM gene (creation date 02/15/2022) and is highly constrained for missense variation (gnomAD Z-score 2.95)." Please change to "... has some constraint for missense variation (gnomAD Z-score 2.95) and is intolerant for loss of function (gnomAD pLI 1.0)."

Minor

Abstract:

- Patient phenotypic spectrum includes developmental delay/intellectual disability, hypotonia, distinct craniofacial features and variable presence of brain, renal, cardiac and musculoskeletal abnormalities > The patient phenotypic spectrum includes developmental delay/intellectual disability, hypotonia, distinct craniofacial features and variable presence of brain, renal, cardiac

and musculoskeletal abnormalities.

- Genotyping analysis supporting true maternity and paternity of families 2 and 5 is available on Table S4 > Genotyping analysis supporting true maternity and paternity of families 2 and 5 is available in Table S4.
- The REVEL scores for this variant is in the tolerable range (0.17), p.L668S is borderline damaging (0.63), and all other variants have predicted damaging consequences. This sentence is awkward. Please reformulate.
- NA (cursif) > NA (not cursif) in Table S1 in Endocrine field for Fam 6.
- Paternally inherited bening > paternally inherited benign in multiple Microarray fields for Fam 6 in Table S1.
- Poorly formulated sentence: The REVEL scores for this variant is in the tolerable range (0.17), p.L668S is borderline damaging (0.63), and all other variants have predicted damaing consequences. Page 8.
- Variable renal abnormalities were noticed in three patients, including unilateral multicystic kidney > Variable renal abnormalities were noticed in three patients, including unilateral multicystic kidney disease.... Page9
- To investigate the protein stability related pathogenicity of WDR44 variants, fibroblasts were isolated from patients.... > To investigate the protein stability related pathogenicity of WDR44 variants, fibroblasts were cultured from patients.... Page 9.
- Immunoblotting analysis demonstrate the protein abundance > Immunoblotting analysis demonstrates the protein abundance... Page 9/10.

Reviewer #2 (Remarks to the Author):

I am pleased to see that the authors have provided additional data and modified the text to take into account most of my comments. I believe the revised version is substantially improved and congratulate the authors for the large amount of work performed by this collaborative effort and on their very nice story.

I have only one minor comment to make: in the discussion (p.22), I feel that the new sentence "Although WDR44 variants have pleiotropic effect with multi-organ dysfunction in several patients, classical ciliopathy-related features may be more subtle" is confusing (in what sense "subtle"?), while also not really addressing my point. Maybe a wording such as: "Overall, our clinical and experimental findings support WDR44 variants as the cause of a pleiotropic disorder that involves many of the organ systems typically involved in ciliopathies through a cilia-linked mechanism. This observed WDR44-related disorder might therefore belong to the continuously expanding list of second-order ciliopathies, etc" could be a good compromise?

Otherwise a few very minor points (typos, grammar, etc):

p.6: I am not sure it is appropriate to design humans as "wildtype"?

p.15: 2 typos: N840S cells uncapping in serum fed and serum starved conditions was not different than controls. Together, these results demonstrate that more severe disease is associated with affected ciliogenesis initiation in patient fibroblast cells.

p.16: the position of the last concluding sentence on the ms variants is a little misplaced after half a paragraph on the truncating variant

p.20: was N840S variant meant? (These results are consistent with the D648G and S764F variants causing more severe ciliopathy-related disease than the N840S patient)

REVIEWERS' COMMENTS

Reviewer #1 (Remarks to the Author):

Thank you for the additional experimentation. The evidence for a gain of function mechanism is still incomplete in my opinion. Prior to acceptance, the claim for gain of function mechanism needs to be significantly weakened throughout the report, particularly in the title (gain-of-function wording should be removed), abstract and discussion. I am not opposed to presenting the gain of function findings as a model for disease mechanism (and I certainly find your findings interesting!), but in my opinion the authors just cannot rule out the possibility of loss of function mechanism and that should be clearly worded in the report. The gain of function hypothesis is a sexy idea, but it is not a common mechanism in ciliopathies and after reading the revision I still feel that I am not convinced. If the authors want to publish their report without additional experimentation they need to clearly acknowledge the possibility that the mechanism of disease could also be loss of function for the following reasons:

As suggested by the reviewer we have toned down the GOF and provide further commentary that LOF throughout the manuscript, in particular with WDR44 p.R733*. Specific instances where this has been done is outlined in response to the reviewers comments below.

We agree that GOF is not a common mechanism but of course this fact does not rule out GOF as a potential ciliopathy mechanism. Our previous work demonstrated that WDR44 acts as a negative regulator of ciliogenesis (Walia et al., 2019 Dev Cell), which was confirmed in this study by knockout of WDR44 in RPE-1 cells (Figure 6b). Based on the observed effects of WDR44 variants on ciliogenesis initiation we believe the GOF is a possible explanation for the ciliopathy-related disease observed in patients.

There is a strong reduction in reduced transcript level of p.R733* in cells derived from this patient (as well as various other patients) and the authors acknowledge there is sensitivity to NMD for this patient in the revised manuscript. There is no evidence shown of protein expression in patient fibroblasts, which could have provided compelling evidence for gain of function effects. The authors only refer to exogenous expression of p.R733*, but I consider that poor evidence; 2) The pLI score of WDR44 is 1 (gnomAD) and suggests this gene does not tolerate loss of function; 3) The 1.9Mb deletion reported by Pavey (PMID: 27194972) involves multiple genes and exon 1 of WDR44 in a patient with an overlapping phenotype. I believe this is a loss of function variant as the first downstream ATG of WDR44 is in exon 3 and this ATG is not in frame. The authors rightfully state that other genes were also deleted in that 1.9Mb deletion and that the evidence for pathogenicity for WDR44 is therefore limited. While that is true, haploinsufficiency of WDR44 is of course suspicious, particularly after reading about the detection of a nonsense variant in your report; 4) The expression of endogenous mutant WDR44 is lower compared to controls in patient-derived fibroblasts (of various individuals with convincing variants anyway) and mutant GFP-tagged WDR44 proteins is also lower compared to wild type WDR44 expression, which is generally consistent with loss of function; 5) The authors did not assess knockout models (fish/fly/mouse) for learning impairment; 6) The phenotype of the p.R733* patient is more severe

compared to missense, which would make sense in context of loss of function.

We have modified and added the following statements to the discussion that we feel addresses the reviewers points . ‘While our studies support a GOF mechanism, we cannot rule out that loss of function (LOF) of WDR44 contributes to the ciliopathy-like phenotype observed, because protein expression data was only available for a limited number of patients. Future experimentation with additional patient cells, as well as assessing neurological function and learning/memory in knockout animal models, will be important for further defining patient disease mechanisms. ‘ As mentioned above we have also modified the text to tone down ‘GOF’ mechanism throughout the manuscript.

Also, please change the title to “Variants in the WDR44 WD40-repeat domain cause syndromic intellectual disability possibly caused by impaired ciliogenesis initiation.”

While we appreciate the reviewers suggestion, we feel the request to tone down the ‘GOF’ mechanism is an appropriate change to the title. We have modified the title and removed ‘Gain-of-function’ from the title and in the abstract. Referencing only ‘syndromic intellectual disability’ in the title, while common to our patients, would not adequately address the other ‘ciliopathy-related’ disease highly prevalent in our patient cohort.

The last sentence of the abstract should be “This study provides new insights into WDR44 WDR structure and characterizes a new syndrome that could result from impaired ciliogenesis. Although a gain of function model has been proposed as a mechanism of disease based on this work, we cannot rule out that loss of function plays a role in the etiology of this disorder at this time.”

We thank the reviewer for this suggestion and have modified the abstract as follows. ‘This study provides new insights into WDR44 WDR structure and characterizes a new syndrome that could result from impaired ciliogenesis.’ As we have removed gain-of-function from the title and abstract we do not think it is necessary to include the last sentence suggested.

In the first paragraph of the discussion, please change to “Based on comprehensive in vitro and in vivo experiments we propose a molecular mechanism of disease wherein WDR44 variants within the WDR result in a gain of function in negative regulation of ciliogenesis through increased RAB11 binding activity. Interestingly, we find that more severe disease causing missense have a destabilizing effect on the variant protein likely resulting from structural defects in the WDR. Moreover, WDR patients variants disrupt interdomain interactions that are associated with regulating binding to RAB11. While this gain of function mechanism is an attractive model, we cannot rule out that loss of function of WDR44 contributes to the ciliopathy-like phenotype in our patients, because [include reasons]”.

We thank the reviewer for this suggestion. We have changed this section as suggest with some modification which we feel appropriately considers the reviewers comments about LOF ‘...While our studies support a GOF mechanism, we cannot rule out that loss of function of WDR44 contributes to the ciliopathy-like phenotype observed, because protein expression data was only available for a limited number of patients. ‘

In the p.R733* discussion, please extend the discussion to “Unfortunately, further studies with R733* protein from patient fibroblasts were not possible at this time and therefore we cannot rule out loss of function as a mechanism of disease”.

We thank the reviewer for this suggestion and have made this change.

In the last paragraph of the discussion, please also address that loss of function (using those words) cannot be ruled out as a mechanism of disease.

We thank the reviewer for this suggestion. However, because we have not addressed GOF mechanism in this final paragraph we felt this change is not necessary as we have referenced LOF earlier in the discussion.

p.R733*

Endogenous protein evaluation in patient-derived fibroblasts would have been very helpful, but I understand that it was not possible despite best efforts. The exogenous expression evidence is weak; please acknowledge that.

It is not clear what the reviewer is referencing. We show p.R733* exogenous expression in human cells and fish. This variant expresses similar to the wild-type protein and is stronger than most missense variants (Figure 2C, 4A, 7A, 7B). In the case of fish expression (Figure 4A) we had noted in the results that expression is lower for all variants compared to wild-type.

Also please adjust this sentence on page 8: “The nonsense variant in exon 16 introduces a stop codon that is predicted to truncate the protein at the residue 733 in WD5 (Figure 1g).”. With the knowledge you now have regarding sensitivity to NMD, I suggest to reword that to “The nonsense variant in exon 16 introduces a premature stop at codon 733 of the encoded protein. This is expected to lead to an absent protein and/or a protein that is truncated at residue 733 in WD5 (Figures 1g, 2b)”.

Thank you for this suggestion. We have modified as suggested.

p.N840S

Thank you for adding a comment relating to the p.N840S variant on page 8 that says “...with the exception of p.N840S that was of uncertain significance”. It is not completely clear from this sentence if the in silico tools render uncertain predictions or if the variant is classified as a variant of uncertain significance. I suggest to reword to “...with the exception of p.N840S, which we classified as a variant of uncertain significance.”. It is important to use clear language as laboratory/clinical geneticists, who will be using your report in the future as a reference, will be looking for variant classifications and whether or not a variant is considered likely causal or not by the authors.

We have modified as suggested.

I also recommend adding a column with classifications in Table S2.

We have added the ACMG classification as suggested.

Pedigrees

Thank you for conducting additional segregation analysis, which has strengthened the genetic portion of your report. Regarding Family 9: please, either remove cousin III:3 from the pedigree as the current figure suggests that ‘the variant does not segregate’ or keep III:3 in the pedigree and grey the square and add a reference in the legend explaining that detailed phenotypic information is lacking but it is thought that the phenotype includes/may include mild speech delay, febrile seizure and ADHD.

We thank reviewer for this suggestion. We have modified the pedigree and the legend of Figure S1 as suggested adding a grey shading for this patient.

Nomenclature

The authors corrected some nomenclature but there are still inconsistencies, e.g. N840S in some sections versus p.N840S in others. Please check all variants (not just N840S) and be consistent where you can, thereby applying HGVS guidelines throughout the manuscript (<https://varnomen.hgvs.org/>).

We thank the reviewer for this suggestion. We have modified to use the nomenclature p.N840S, for example, when referring to patient and patient cell data. Where exogenous proteins are expressed or examined we have omitted the use of ‘p.’ as these are not from patient source.

Variant Interpretation

Thank you for adding the Table S3 with variant prioritization data for family 1. While I appreciate the addition, the table is not very easy to review in its current form. It would help to have information such as RefSeq/cDNA/protein nomenclature and zygosity in the start of the table (rather than somewhere in the middle).

We moved the relevant columns of RefSeq/cDNA/protein at the beginning of the file excel. We also added a column for variant interpretation and prioritization and ACMG classification when applicable (for known disease causing genes)

I also suggest to add a column with brief comments for each variant in the start of Table S3 why it was (de-)prioritized (and possibly ACMG classification). The authors claim they reviewed GTEX expression, protein atlas database and literature but that information is not provided.

The variant(s) of interest (mainly WDR44) were manually inspected in GTEX after applying the filtering and prioritization strategies, which are now explained in Table S3.

Also regarding variant prioritization, the authors state in their revised report “A comprehensive analysis looking at all genes and considering all possible mechanisms of inheritance did not identify other candidate variants.” I think that may be formulated too strong. The authors should consider changing that to “A comprehensive analysis looking at all genes and considering all possible mechanisms of inheritance did not identify stronger candidate variants.”

We thank the reviewer for this suggestion and we have modified as suggested.

The following sentence is also too strong “WDR44 was recently listed as an OMIM gene (creation date 02/15/2022) and is highly constrained for missense variation (gnomAD Z-score 2.95).” Please change to “... has some constraint for missense variation (gnomAD Z-score 2.95) and is intolerant for loss of function (gnomAD pLI 1.0).”

We thank the reviewer for this suggestion and have modified as suggested.

Minor

Abstract:

- Patient phenotypic spectrum includes developmental delay/intellectual disability, hypotonia, distinct craniofacial features and variable presence of brain, renal, cardiac and musculoskeletal abnormalities > The patient phenotypic spectrum includes developmental delay/intellectual disability, hypotonia, distinct craniofacial features and variable presence of brain, renal, cardiac and musculoskeletal abnormalities.

We thank the reviewer for this suggestion and have modified as suggested.

- Genotyping analysis supporting true maternity and paternity of families 2 and 5 is available on Table S4 > Genotyping analysis supporting true maternity and paternity of families 2 and 5 is available in Table S4.

Thank you. Modified as suggested

- The REVEL scores for this variant is in the tolerable range (0.17), p.L668S is borderline damaging (0.63), and all other variants have predicted damaging consequences. This sentence is awkward. Please reformulate.

We appreciate this suggestion and have changed to the following. ‘The REVEL score for this variant falls within the tolerable range (0.17), p.L668S is borderline damaging (0.63), and all other variants are predicted to have damaging consequences.’

- NA (cursif) > NA (not cursif) in Table S1 in Endocrine field for Fam 6.

Thank you we have modified.

- Paternally inherited benign > paternally inherited benign in multiple Microarray fields for Fam 6 in Table S1.

Thank you we have modified.

- Poorly formulated sentence: The REVEL scores for this variant is in the tolerable range (0.17), p.L668S is borderline damaging (0.63), and all other variants have predicted damaging consequences. Page 8.

Modified as described above

- Variable renal abnormalities were noticed in three patients, including unilateral multicystic kidney > Variable renal abnormalities were noticed in three patients, including unilateral multicystic kidney disease.... Page9

Thank you for suggestion. We have modified as suggested.

- To investigate the protein stability related pathogenicity of WDR44 variants, fibroblasts were isolated from patients.... > To investigate the protein stability related pathogenicity of WDR44 variants, fibroblasts were cultured from patients.... Page 9.

Thank you for suggestion. We have modified as suggested

- Immunoblotting analysis demonstrate the protein abundance > Immunoblotting analysis demonstrates the protein abundance... Page 9/10.

Thank you for suggestion. We have modified as suggested

Reviewer #2 (Remarks to the Author):

I am pleased to see that the authors have provided additional data and modified the text to take into account most of my comments. I believe the revised version is substantially improved and congratulate the authors for the large amount of work performed by this collaborative effort and on their very nice story.

I have only one minor comment to make: in the discussion (p.22), I feel that the new sentence "Although WDR44 variants have pleiotropic effect with multi-organ dysfunction in several patients, classical ciliopathy-related features may be more subtle" is confusing (in what sense "subtle"?), while also not really addressing my point. Maybe a wording such as: "Overall, our clinical and experimental findings support WDR44 variants as the cause of a pleiotropic disorder that involves many of the organ systems typically involved in ciliopathies through a cilia-linked mechanism. This observed WDR44-related disorder might therefore belong to the continuously expanding list of second-order ciliopathies, etc" could be a good compromise?

We thank the reviewer for this suggestion and have made the suggested modification.

Otherwise a few very minor points (typos, grammar, etc):
p.6: I am not sure it is appropriate to design humans as "wildtype"?

Thank you for this observation. We have changed to "unaffected".

p.15: 2 typos: N840S cells uncapping in serum fed and serum starved conditions was not different than controls. Together, these results demonstrate that more severe disease is associated with affected ciliogenesis initiation in patient fibroblast cells.

Correction made, thank you.

p.16: the position of the last concluding sentence on the ms variants is a little misplaced after half a paragraph on the truncating variant

Thank you for the suggestion, we have changed position of this sentence.

p.20: was N840S variant meant? (These results are consistent with the D648G and S764F variants causing more severe ciliopathy-related disease than the N840S patient)

We thank the reviewer for pointing this out. We have changed the sentence to “These results are consistent with the p.D648G and p.S764F variants causing ciliopathy-related disease.”